

# Relating optical and microwave grain metrics of snow: The relevance of grain shape

Quirine Krol[1] and Henning Löwe[1]

[1]WSL Institute for Snow and Avalanche Research SLF, Flüelastrasse 11, 7260 Davos Dorf, Switzerland

*Correspondence to:* Henning Löwe (loewe@slf.ch)

**Abstract.** While optical properties of snow are predominantly determined by the specific surface area (SSA), microwave measurements are often analyzed in terms of the exponential correlation length $\xi$. A statistical relation between both is commonly employed to facilitate forcing of microwave models by optical measurements. To improve the understanding of $\xi$ and establish a link between optical and microwave grain metrics we analyzed the third order term in the expansion of the correlation function that can be regarded as a shape parameter related to mean and Gaussian curvature. We show that the statistical prediction of the correlation length via SSA is considerably improved by including the shape metric. In a second step we address the chord-length distribution as a key quantity for geometrical optics. We show that the second moment of the distribution can be accurately related to density, SSA and the shape parameter. This empirical finding is supported by a theoretical relation between the chord length distribution and the correlation function as suggested by small angle scattering methods. As a practical implication, we compute the optical shape factor $B$ from tomography data. Our results indicate a possibility of estimating $\xi$ by a careful analysis of shape corrections in geometrical optics.

## 1 Introduction

Linking physical properties and microstructure of snow is a fundamental task of snow science. The two-point correlation function of snow has become a key quantity in this respect for the prediction of various properties such as thermal conductivity, permeability and electromagnetic properties of snow (Wiesmann and Mätzler, 1999; Löwe et al., 2013; Calonne et al., 2014b; Löwe and Picard, 2015). The recent gain in interest of correlation functions is mainly driven by available data from micro-computed tomography ($\mu$CT), from which the correlation function can be conveniently estimated. The analysis of correlation functions for microwave application dates back to the pre-$\mu$CT era, where thin section data and stereology were used to obtain the required information (Vallese and Kong, 1981; Zurk et al., 1997; Mätzler and Wiesmann, 1999).

The relevance of the two-point correlation function for microwave modeling originates from the connection between its Fourier transform and the scattering phase function in the Born approximation for small scatterers (Mätzler, 1998; Ding et al., 2010; Löwe and Picard, 2015), or the connection to the effective dielectric tensor via depolarization factors (Leinss et al., 2015). A common way to characterize the correlation function is a fit to an exponential, such that the fit parameter, the so called exponential correlation length $\xi$, can be used to model the decay of structural correlations in snow. This approach dates back to Debye et al. (1957) in the context of small angle scattering of heterogeneous materials. However the characterization



of snow in terms of a single size metric $\xi$ is only an approximation since the occurence of multiple length scales (Löwe et al., 2011) are known to play a role in anisotropy (Löwe et al., 2013; Calonne et al., 2014b). Despite these fundamental caveats, the correlation length $\xi$ still constitutes the main microstructural parameter for microwave modeling of snow (Proksch et al., 2015a; Pan et al., 2016) if the Microwave Emission Model of layered snowpacks (Wiesmann et al., 1998) is used. However,

direct measurements of $\xi$, besides $\mu$CT, do not exist and the correlation length is often statistically inferred from measurements of the optical equivalent diameter $d_{\mathrm{opt}}$ or of specific surface area (SSA). This link was established statistically (Mätzler, 2002) leading to the empirical relation

$$\xi \approx 0.5 d_{\mathrm{opt}}(1 - \phi), \tag{1}$$

where $\phi$ is the ice volume fraction. This relation facilitates the use of the measured optical diameter as the primary input for

microwave modeling (Durand et al., 2008; Proksch et al., 2015b; Tan et al., 2015). Despite this practical advantage, such a relation can only serve as a first approximation, since the prefactor in Eq. (1) seems to depend on snow type (Mätzler, 2002), causing significant scatter in the estimates. This has neither been investigated in detail nor traced back to additional shape metrics.

A similar issue of shape, though less significant in order of magnitude, emerges in the context of optical measurements.

Optical properties (e.g. reflectance) can be largely predicted from the optical diameter or SSA (Kokhanovsky and Zege, 2004). The remaining scatter is small but commonly also attributed to grain shape. The influence of shape on light penetration was recently quantified by Libois et al. (2013) in terms of a shape factor $B$, which originates from Kokhanovsky and Zege (2004). A systematic framework that principally allows to analyze this issue for geometrical optics was recently put forward by Malinka (2014) who derived closed-form expressions for the averaged optical properties. The relevant microstructural quantity is the

chord length distribution (Torquato, 2002) or, more precisely, its Laplace transform. Thereby, the microstructural metrics used by Malinka (2014), is not limited to a particular model microstructure (e.g, spheres) but can be applied to generic two-phase media which implicitly incorporates shape.

A key requirement for potential shape metrics is a well-defined geometrical meaning of the quantity. Presently, the exponential correlation length is essentially a statistical object which is still difficult to interpret beyond the empirical correlation in

Eq. (1). This hinders the development of evolution equations in snowpack models, and the development of alternative, portable measurement techniques to estimate new parameters in the field for validation campaigns. Present snowpack models (Vionnet et al., 2012; Lehning et al., 2002) contain empirical shape descriptors such as sphericity (Brun et al., 1992). An objective definition of these quantities for arbitrary two-phase materials is, however, not possible. New shape metrics should thus ideally seek to replace empirical microstructure parameters by an objective, measurable and geometrically comprehensible metric

of the microstructure. An appealing candidate is a curvatures based metric, because i) curvatures have already been used to comprehend snow metamorphism via mean and Gaussian curvatures (Brzoska et al., 2008; Schleef et al., 2014; Calonne et al., 2014a) ii) curvatures are natural to assess shape via deviations from a sphere, very close to the original idea of sphericity (Brun et al., 1992) and iii) curvatures also emerge as higher order terms in the expansion of the correlation function (Torquato, 2002), which closes the circle with the microwave context.





The motivation of the present paper is three-fold. First, we will systematically assess the curvature term in the expansion of the correlation function as a potential shape parameter. We will be guided by the question if and how the well-known statistical relation Eq. (1) between the exponential correlation length and the optical diameter can be improved by incorporating curvatures. Second, we will characterize the microstructure in terms of chord length distributions in order to make contact to aspects of shape in snow optics. Third, we motivate an approximate relation between the correlation function and the chord length distribution that was suggested in the context of small angle scattering (Méring and Tchoubar, 1968). The relation suggests various connections between the moments of the chord length distributions, surface areas, curvatures and the exponential correlation length. The statistical analysis of these metric inter-relations leads to the announced microstructural connection between geometrical optics and microwave scattering in the Born approximation, and an expression for the optical shape factor $B$.

In Section 2 we present the theoretical background for the correlation function, the chord length distribution, the relation between both quantities and the governing length scales. In Section 3 we provide a summary of the image analysis methods. To provide confidence of the interpretation of the curvature metrics from the correlation function we present an independent validation of these quantities via the triangulation of the ice-air interface. The results of the statistical models are presented in Section 4 and discussed in Section 5. Due to the differences in lengthscales between optical and microwave metrics a connection between the two via shape may seem surprising. We therefore aim to illustrate this connection by discussing it in view of the appealing but limited picture of snow as a packing of irregularly shaped grains.

## 2 Theoretical background

### 2.1 Two-point correlation function and microwave metrics

The interaction of microwaves with snow are commonly interpreted as scattering at permittivity fluctuations in the microstructure. This is reflected for example by the fact that in the Born approximation the scattering coefficient or the phase matrix is proportional to the Fourier transform of the two-point correlation function (Mätzler, 1998; Ding et al., 2010; Löwe and Picard, 2015). The correlation function can be derived from spatial distribution of ice and air that is characterized by the ice phase indicator function $\mathcal{I}(\boldsymbol{x})$, which is equal to 1 for a point $\boldsymbol{x}$ in ice and 0 for $\boldsymbol{x}$ in air. From that, a covariance function can be defined which is often referred to as the correlation function

$$C(\boldsymbol{r}) = \overline{\mathcal{I}(\boldsymbol{x} + \boldsymbol{r})\mathcal{I}(\boldsymbol{x})} - \phi^2. \tag{2}$$

In the following we disregard anisotropy by stating that $C(r)$ only depends on the magnitude of $r = |\boldsymbol{r}|$. To interpret snow with this approach, an average over different coordinate directions must be carried out.

The value of the correlation function $C(0) = \phi(1 - \phi)$ is simply related to the volume fractions of ice and air. Therefore, often only the normalized correlation function

$$A(r) = C(r)/C(0) \tag{3}$$





is used, (see Fig. 1b). Since $A(r)$ must decay from $A(0) = 1$ to zero for $r \to \infty$, the correlation function is often described by an exponential form

$$A(r) = \exp\left(-r/\xi\right), \tag{4}$$

in terms of a single length scale, the exponential correlation length $\xi$, which empirically characterizes the decay of $A(r)$.

In contrast, for small arguments $r$, also rigorous results for the correlation can be inferred since the expansion of $A(r)$ can be interpreted in terms of geometrical properties of the interface. According to Torquato (2002), the expansion for an isotropic medium reads

$$A(\boldsymbol{r}) = 1 - \frac{r}{\lambda_1}\left[1 - \frac{r^2}{\lambda_2^2} + \mathcal{O}(r^3)\right] \tag{5}$$

in terms of the length scales $\lambda_1, \lambda_2$. The first order term

$$\frac{1}{\lambda_1} = -\left.\frac{d}{dr}A(r)\right|_{r=0} = \frac{s}{4\phi(1-\phi)}, \tag{6}$$

is the slope of the correlation function at the origin and can be expressed in terms of $s$ which is the interfacial area per unit volume (Debye et al., 1957). The size metric $\lambda_1$ is one of the most fundamental lengths scales for a two-phase medium and commonly referred to as the Porod length in small angle scattering, or simply correlation length in Mätzler (2002). The metric $\lambda_1$ can be also related to the SSA, defined as the surface area per ice mass $(\mathrm{m}^2/\mathrm{kg})$, or in turn to the equivalent optical diameter

$d_{\mathrm{opt}}$ of snow via

$$\lambda_1 = \frac{4\phi(1-\phi)}{s} = \frac{4(1-\phi)}{\rho_{\mathrm{i}}\,\mathrm{SSA}} = \frac{2(1-\phi)}{3}\,d_{\mathrm{opt}} \tag{7}$$

with $\rho_{\mathrm{i}}$ representing the density of ice. The last equality is obtained when the definition of $d_{\mathrm{opt}} = 6/\rho_i\mathrm{SSA}$ is inserted (see Mätzler (2002)). For a two-phase material with a smooth interface, the second order term $\sim r^2$ is missing in the expansion Eq. (5) and the next non-zero term is the cubic one with a prefactor $1/\lambda_1\lambda_2^2$. Here the length scale $\lambda_2$ also has a geometric

interpretation in terms of interfacial curvatures, hereafter referred to as the curvature length. As originally shown by Frisch and Stillinger (1963), the following identity holds

$$\frac{1}{\lambda_2^2} = \lambda_1\left.\frac{d^3}{dr^3}A(r)\right|_{r=0} = \frac{\overline{H^2}}{8} - \frac{\overline{K}}{24} \tag{8}$$

in terms average squared mean curvature $\overline{H^2}$ and the averaged Gaussian curvature $\overline{K}$. The quantity $\lambda_2^{-2}$ it also referred to as Eulerian curvature of an interface (Tomita, 1986). The averaged Gaussian curvature $\overline{K}$ is related to a topological quantity of

the ice-air interface. It can be related to the Euler characteristic $\chi$ via the Gauss–Bonnet theorem

$$\chi = \frac{1}{2\pi}\int d^2x\, K(x) = Vs\overline{K}, \tag{9}$$

with $V$ representing the total volume. This is noteworthy insofar, as the (local) expansion of the correlation function at the origin contains a topological (i.e. a global) property of the interface.





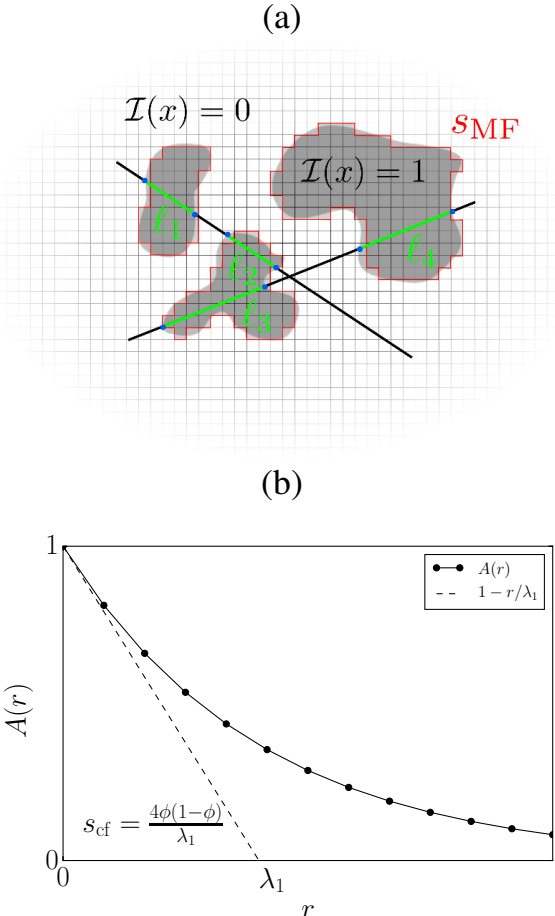

**Figure 1.** a) Illustration of the chord lengths obtained from an ice sample. The mean chord length is defined as the average length of the green line lengths. A stereological approach (Underwood, 1969) to calculate $s$ is to count the number of blue dots per unit length. The estimation for $s_{\mathrm{mf}}$ is given by the red contour. b) Illustration of the correlation function $A(r)$ and the method obtaining an estimate for $s_{\mathrm{cf}}$ by fitting the slope at the origin.

## 2.2 Chord length distributions and optical metrics

In contrast to the interaction with microwaves, snow optics is based on a different microstructural characterization within radiative transfer theory (Kokhanovsky and Zege, 2004), which commonly employs a single metric, the optical diameter. An interesting extension for geometrical optics in arbitrary two-phase media was recently put forward by Malinka (2014). Thereby, the microstructure is taken into account by the chord length distribution of medium which can be unambiguously defined for arbitrary two-phase random media (Torquato, 2002).



Chord lengths in an isotropic medium can be defined as the lengths of the intersections of random rays through the sample with the ice phase as shown in the schematic in Fig. 1a. The chord length distribution $p(\ell)$ of the ice phase denotes the probability (density) for finding a chord of length $\ell$.

In contrast to the Born approximation for microwaves, where the microstructure enters as the Fourier transform of the correlation function, the theoretical approach Malinka (2014) relates the key optical quantities (absorption, phase function, asymmetry-factor) to the Laplace transform of the chord length distribution $p(\ell)$ which is denoted by

$$\widehat{p}(z) = \int_0^\infty d\ell\, p(\ell) e^{-z\ell} \qquad (10)$$

with Laplace variable $z$. The Laplace transform is closely related to the moments of the chord length distribution

$$\mu_n = \int_0^\infty d\ell\, \ell^n p(\ell) \qquad (11)$$

since the expansion of the Laplace transform Eq. (10) for small $z$ can be written as

$$\widehat{p}(z) = 1 - \mu_1 z + \frac{\mu_2}{2} z^2 + \mathcal{O}(z^3). \qquad (12)$$

This implies that the optical response of snow can be systematically improved by successively including higher moments of the chord length distribution. According to theory of Malinka (2014), the Laplace transform has to be evaluated for $z = \alpha$ with the absorption coefficient $\alpha = 2\pi\kappa/\lambda$. Here $\lambda$ is the wavelength and $\kappa$ the imaginary part of the refractive index of ice. It is generally sufficient (Malinka, 2014) to retain only a few terms in Eq. (12). It is straightforward to show (Underwood, 1969) that the first moment, i.e, the mean chord length $\mu_1$ is given by

$$\mu_1 = \frac{4\phi}{s} = \frac{\lambda_1}{1-\phi} \qquad (13)$$

and thus related to the surface area per unit volume $s$ from Eq. (6) or one of its counterparts via Eq. (7). Thus, in lowest order, the Laplace transform Eq. (10) only contains the optical radius or specific surface area of snow. The next order correction involves the second moment $\mu_2$ for which no geometric interpretation has been hitherto given for arbitrary two-phase random media.

The chord length distribution is closely related to stereological principles which have been widely used in the pre-$\mu$CT era (e.g, Buser and Good (1987); Good (1989)), to estimate the density and the surface area per unit volume for snow and other crystalline materials. The connection to stereology is illustrated in Fig. 1a, where the well-known counting of the blue intersection points per unit length gives an estimate for the averaged interfacial area $s$.

## 2.3 Connection between chord lengths and correlation lengths

Following the previous two sections, a link between optical and microwave metrics of snow thus requires to establish a link between correlation functions and chord length distributions. This issue has been discussed by Roberts and Torquato (1999),





who established an exact relation between the Laplace transforms of i) the correlation function, ii) the chord length distribution, and iii) the surface-void correlation function (Torquato, 2002). Despite the apparent complexity, the approach in Roberts and Torquato (1999) still involves the simplified assumption that consecutive chords on the random ray in Fig. 1a are statistically independent. Though this assumption is never strictly met, it is shown in Roberts and Torquato (1999) that this is not a practical

limitation. Their relation also provides a very good approximation for correlated structures such as bicontinuous Gaussian random fields, but at the expense of the complexity from the numerical inversion of Laplace transforms.

To this end we start from a yet simpler relation between the correlation function and chord length distribution that was put forward in the early stages of small angle scattering (Méring and Tchoubar, 1968) to interpret the scattering curve in terms of particle properties. In the present notation the relation can be written as

$$p(\ell) = \mu_1 \frac{d^2}{d\ell^2} A(\ell),\tag{14}$$

which was also used by Gille (2000). The equation was derived for dilute assemblies of convex particles, an assumption which is not valid for snow. However, Eq. (14) has already some non-trivial implications which can be used for the subsequent analysis.

As a first consistency check of the approximation Eq. (14), we can compute the first moment of the chord length distribution

from Eq. (11) for $n = 1$, by inserting Eq. (14) and integrating by parts. This yields $\mu_1 = \mu_1 A(0)$ which is correct by virtue of Eq. (3).

As a next step, we aim at an expression for the second moment of the chord length distribution in terms of interfacial curvatures according to Eq. (11) for $n = 2$. Again, inserting Eq. (14) and integrating by parts yields

$$\mu_2 = 2\mu_1 \int\limits_0^\infty A(r) = 2\mu_1 \lambda_1 f\left(\frac{\lambda_2}{\lambda_1}\right)\tag{15}$$

with an unknown scaling function $f$. To motivate the second equality in (15) we note that the expansion (5) implies that $A(r)$ depends at least on two independent length scales, $\lambda_1$ and $\lambda_2$. As a dimensionless quantity, $A(r)$ can only depend on (arbitrarily chosen) ratios of involves length scales. In the absence of other relevant scales, the correlation function must have the form $A(r) = A(r/\lambda_1, \lambda_2/\lambda_1)$. In turn, the integral over $A(r)$ in (15) has units of length and must have the form $\int_0^\infty A(r) = \lambda_1 f(\lambda_2/\lambda_2)$ with an unknown function $f$. The representation (15) is thus an implication of dimensional analysis.

The validity of the main relation for the chord length distribution Eq. (14) can be assessed by experimental data and the inferred connection Eq. (15) between the second moment of the chord length distribution and interfacial curvatures will guide us in retrieving an empirical relation for the second moment $\mu_2$ in terms of shape.





## 3   Methods

### 3.1   Data

For the following analysis we used an existing dataset of microstructures reconstructed by $\mu$CT previously used in Löwe et al. (2013) for thermal conductivity analysis and Löwe and Picard (2015) for a comparison of microwave scattering coefficients.

All samples were classified according to Fierz et al. (2009) as described in the supplement of Löwe et al. (2013).

### 3.2   Geometry from correlation functions

Obtaining the normalized correlation function $A(r)$ from a $\mu$CT image can be conveniently done by using the Fast Fourier Transform (FFT) as e.g. described in Newman and Barkema (1999). The FFT is typically used for performance issues to evaluate the convolution integral Eq. (2) since direct methods can be very slow. The spatial resolution of the correlation

function depends on the voxel size $\Delta$ which ranges from $18$ to $50$ $\mu$m. The normalized correlation function is obtained in the $x, y$ and $z$ direction and averaged arithmetically over these three directions i.e, $A(r) = (A_x(r) + A_y(r) + A_z(r))/3$, to average out anisotropy.

From the normalized correlation function two types of parameter fittings are performed. First, the exponential correlation length $\xi$ is obtained by fitting the $\mu$CT data to the exponential form Eq. (4). Technically, we estimated the inverse parameter

$k$ by least-squares optimization of the model $A(r) = \exp(-kr)$ to the data in a fixed range of $0 < r < 50\Delta$. An illustration of this method is shown in Fig. 1b. In the following we denote by $\xi$ the inverse of the optimal fit parameter $\xi := 1/k$. Second, we estimated the expansion parameters $\lambda_1$ and $\lambda_2$ of the correlation function by a least-squares regression to the expansion Eq. (5). Technically, we fitted $A(r) = 1 - k_1 r(1 - k_2 r^2)$ in the fixed range of $0 < r < 3\Delta$ which determines the derivatives at the origin. In the following we denote $\lambda_1^{\mathrm{cf}}$ and $\lambda_2^{\mathrm{cf}}$ by the inverse of the optimal fit parameters $\lambda_1^{\mathrm{cf}} := 1/k_1$ and $\lambda_2^{\mathrm{cf}} := 1/k_2$.

The superscript is added to discern these correlation function based estimates from those presented in the next section for a validation.

### 3.3   Geometry from triangulations

Essential for the present analysis in view of shape is the geometrical interpretation (Eq. (6) and Eq. (8)) of the parameters $\lambda_1^{\mathrm{cf}}$ and $\lambda_2^{\mathrm{cf}}$ obtained from the correlation function. To confirm this interpretation, and to make contact of the present method to previous work on curvature properties of the ice-air interface, we also compute these parameters by independent means.

previous work on curvature properties of the ice-air interface, we also compute these parameters by independent means.

To this end we provide alternative estimates $\lambda_1^{\mathrm{vtk}}$ and $\lambda_2^{\mathrm{vtk}}$ from a VTK-based image analysis (www.vtk.org) yielding estimates of the surface area and local curvatures via triangulation as described in Krol and Löwe (2016). In short, a triangulated ice-air interface is obtained by applying a VTKContour filter. After this step, the interface still resembles the underlying voxel structure. Therefore, in a second step the triangulated interface is smoothed by applying the VTKSmoothing filter which in-

volves a smoothing parameter $S$. For further details see Krol and Löwe (2016).





### 3.4 Accuracy of surface area and curvatures estimates

The measured total surface area is obtained by integrating (summing) the surface area of the triangles over the surface and the estimate $\lambda_1^{\mathrm{vtk}}$ naturally depends on the smoothing parameter. A comparison of the triangulation and the correlation function based length scale is shown in Fig. 2 (middle row). A higher value of the smoothing parameter implies a lower surface area $s$

(caused by shrinking of the enclosed volume upon smoothing) and in turn higher estimates for $\lambda_1^{\mathrm{vtk}}$. It is illustrative to show that even without smoothing for $S = 0$ the obtained triangulated surface is still different from the voxel surface $s_{\mathrm{mf}}$, which is obtained by the union of ice-air transition faces in the voxel based image (as illustrated by the red contour in Fig. 1a). The quantity $s_{\mathrm{mf}}$ is one of the four Minkowski functionals and can be computed by standard counting algorithms (Michielsen and Raedt, 2001). For isotropic systems, and statistically representative samples, the relation between the surface obtained from the

correlation function $s_{\mathrm{cf}} = 4\phi(1-\phi)/\lambda_1^{\mathrm{cf}}$ and the Minkowski functionals is known to be $s_{\mathrm{cf}} = 2s_{\mathrm{mf}}/3$ as discussed in Torquato (2002, p. 290) and shown here in Fig. 3.

An estimate for the curvature length $\lambda_2^{\mathrm{vtk}}$ is obtained from the VTKCurvature filter on the triangulated ice-air interface yielding local values for mean and Gaussian curvature which can be integrated to compute $\lambda_2^{\mathrm{vtk}}$ via Eq. (8). The comparison of the triangulation based curvature length and the correlation function based curvature length is shown in Fig. 2 (bottom row).

The parameters $\lambda_1^{\mathrm{vtk}}$ and $\lambda_2^{\mathrm{vtk}}$ depend strongly on the smoothing parameter $S$. The value $S = 200$ performed best by comparing the value $\lambda_2^{\mathrm{vtk}}$ to $\lambda_2^{\mathrm{vtk}}$, see Fig. 2 (bottom row).

Overall, the comparison provides reasonable confidence that the geometrical interpretation of the correlation function parameters is correct, though uncertainties inherent to the smoothing operations must be acknowledged. In the following we solely use the quantities derived from the correlation function, viz. $\lambda_1 = \lambda_1^{\mathrm{cf}}$ and $\lambda_2 = \lambda_2^{\mathrm{cf}}$ where the superscripts are omitted

for brevity.

### 3.5 Chord length distribution

To compute the ice chord length distribution from the binary images, *all* linear lines through the sample in all three Cartesian directions $\beta = x, y, z$ are considered and *all* ice chords were measured and binned to obtain direction dependent counting densities $n^\beta(\ell)$. Here $n^x(\ell)$ denotes the total number of chords in $x$ direction which have length $\ell$. For a binary CT image, $\ell$

can take integer values $0 < \ell < L_x$ which are restricted by the sample size $L_x = N_x\Delta$ and the voxel size $\Delta$ of the image. The mean chord length and other moments $\mu_i$ are then computed from

$$\mu_i = \frac{1}{\sum_{\ell,\beta} n^\alpha(\ell)} \sum_{\ell,\beta} \ell^i n^\beta(\ell) \tag{16}$$

### 3.6 Statistical models

A main part of the following analysis comprises statistical relations between the length scales derived from the chord length

distribution and the correlation function in section 2. In total, we will consider a few statistical models that first relate the exponential correlation length $\xi$ and $\mu_2$ to the geometrical length scales $\lambda_1$ and $\lambda_2$ and second, relate $\xi$ to $\mu_1$ and $\mu_2$. We will





**Figure 2.** Comparison between smoothing parameter $S = 50$ (left) and $S = 200$ (right) for the top: Representation of the surface of a subsection of a snow sample. In the middle: Scatter plots of the correlation length $\lambda_1^{\mathrm{cf}}$ versus $\lambda_1^{\mathrm{vtk}}$, including a fit (red dotted line). At the bottom: Scatter plots of the curvature length $\lambda_2^{\mathrm{cf}}$ versus $\lambda_2^{\mathrm{vtk}}$, including a fit (red dotted line).



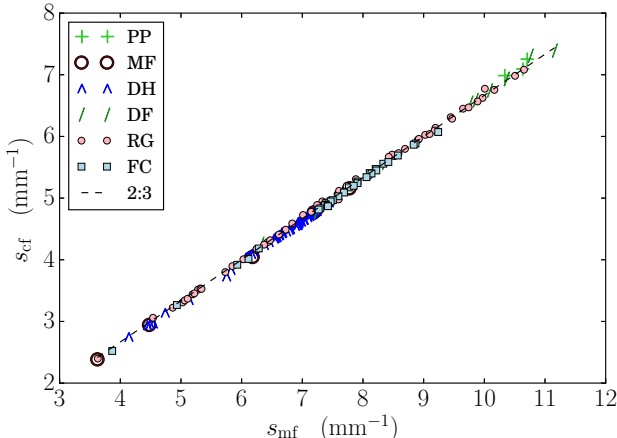

**Figure 3.** Scatter plot of the averaged interfacial area obtained by the the correlation function method $s_{\mathrm{cf}}$ versus Minkowski functionals method $s_{\mathrm{mf}}$.

start with a one-parameter statistical model and compare the results to the two parameter models. We will assess the quality of the fits with the correlation coefficient $R^2$.

## 4  Results

### 4.1  Relating exponential correlation length to optical diameter

As a starting point for the statistical analysis we revisit the empirical relation

$$\xi = 0.75\lambda_1, \tag{17}$$

which is equivalent to Eq. (1) by virtue of Eq. (7), as suggested by Mätzler (2002). To this end we fitted $\xi$ and $\lambda_1$ and obtained an average slope of 0.79 with a correlation coefficient of $R^2 = 0.733$, shown by the green dashed line in Fig. 4a. In the next step we fitted the same data to include an intercept parameter

$$\xi = a_0 + a_1\lambda_1. \tag{18}$$

Here the correlation coefficient is $R^2 = 0.731$ and and the parameters are given by $a_0 = 5.93 \times 10^{-2}$ mm, $a_1 = 0.794$, with very low $p$-values ($p < 5 \times 10^{-4}$) for the intercept and the slope ensuring the significance of the parameters of the fit. The order of magnitude of the intercept $a_0$ is negligible.





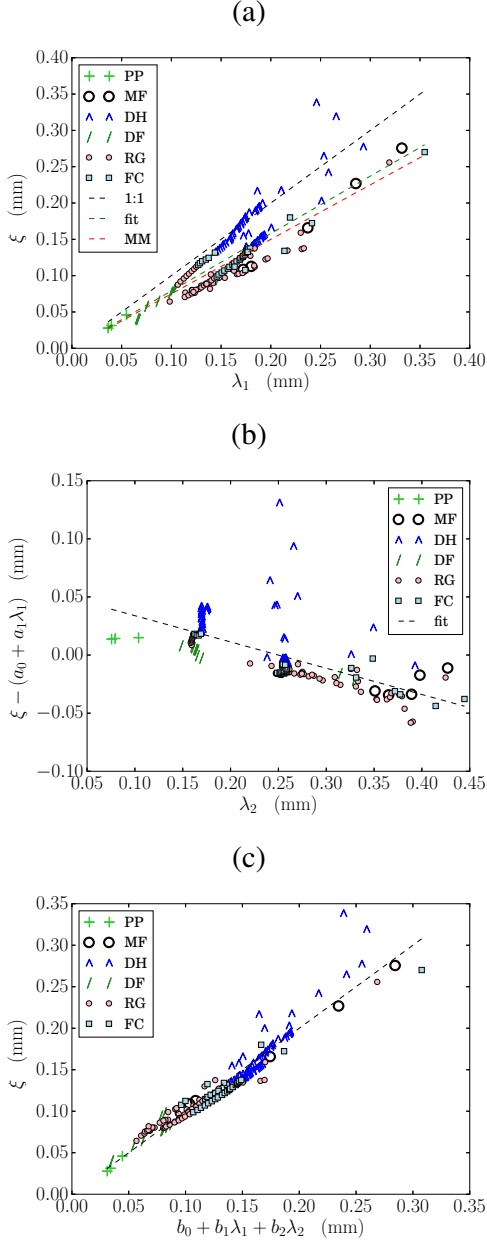

**Figure 4.** Scatter plots of a) the exponential correlation length $\xi$ versus the correlation length $\lambda_1$. A linear fit is plotted in green. Additionally the prediction of Eq. (17) (MM) is plotted in red. b) The residuals of $\xi$ and the statistical model Eq. (18), versus the curvature length $\lambda_2$. c) The statistical model Eq. (19) predicting $\xi$ depending on the optical diameter $\lambda_1$ and the curvature length $\lambda_2$.





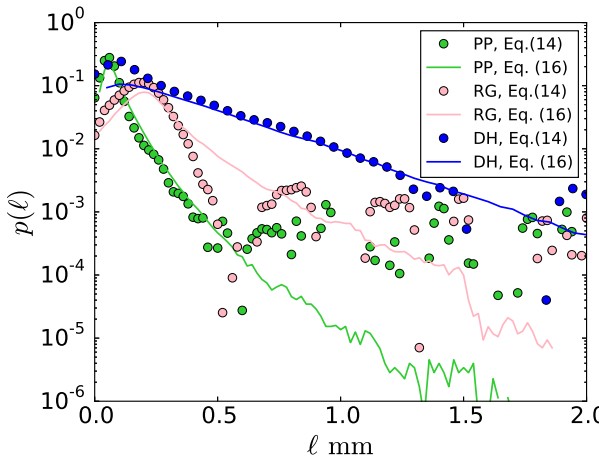

**Figure 5.** Plot of the chord length distributions computed by Eq. (14) (symbols) and by direct analysis, Eq. (16) (solid-line).

## 4.2 Relating the exponential correlation length to the correlation length and curvature length

As a next step we have included the curvature length $\lambda_2$ and fitted the exponential correlation length $\xi$ to the model

$$\xi = b_0 + b_1\lambda_1 + b_2\lambda_2 \tag{19}$$

The results are shown in Fig. 4c. Here we find an improvement compared to Eq. (17). The correlation coefficient is $R^2 = 0.922$

and the fit parameters are given by $b_0 = 1.23 \times 10^{-2}$ mm, $b_1 = 1.32$ and $b_2 = -3.85 \times 10^{-1}$. The $p$-values are very small for all coefficients $b_i$. The order of magnitude of the improvement can already be roughly estimated from the ratio of the prefactors $b_1$ and $b_2$. To provide further evidence that the improvement of the prediction comes from the curvature length, we analyzed the residuals of the prediction Eq. (18) and plotted $\xi - (a_0 + a_1\lambda_1)$ versus the curvature length scale $\lambda_2$ as shown in Fig. 4b.

The residuals of $\xi$ with the statistical model Eq. (18) show a correlation with $\lambda_2$ of $R^2 = 0.644$, which eventually causes the

improvement for the exponential correlation length.

## 4.3 Connection between chord length distributions and correlation functions

To bridge to the chord length metrics, we first assess the relation between the chord length distribution $p(\ell)$ and the correlation function $A(\ell)$ as suggested by Eq. (16). To this end we compared the chord length distribution obtained directly from the $\mu$CT image (cf. section 3.5) with the prediction of Eq. (16) via the correlation function for a few examples of different snow

types. The results are shown in Fig. 5. The selected snow samples are the same as those used in Löwe and Picard (2015, Fig. 8 and Fig. 9). Qualitatively, the characteristic form (i.e, single maximum), the location of the maximum, and the width of the distribution are correctly predicted by Eq. (16). On the other hand, there are obvious shortcomings, such as the oscillatory tail for the RG example when the chord length distribution is derived via Eq. (16). We will revisit this feature in the discussion.





### 4.4 Second moment of the chord length distribution

Using the previous results we can derive an approximate relation between the second moment of the chord length distribution and the interfacial curvatures. To motivate a statistical model we build on Eq. (14), which suggests a general scaling form

$$\frac{\mu_2}{2\mu_1} = \lambda_1 f\left(\frac{\lambda_2}{\lambda_1}\right). \tag{20}$$

5 We investigate the validity of this expression by approximating the unknown function $f$ by successively higher orders of $\lambda_2/\lambda_1$ in a statistical model. In a first step we approximate $f$ by a constant using the statistical model

$$\frac{\mu_2}{2\mu_1} = l_0 + l_1\lambda_1. \tag{21}$$

Although not predicted by Eq. (20), we again allow for an interception term $l_0$ similar to Eq. (18), and Eq. (19). The optimal parameters for the model Eq. (21) are $l_0 = -2.40 \times 10^{-2}$ mm and $l_1 = 1.25$, with negligible $p-$values and a correlation 10 coefficient of $R^2 = 0.898$. The results are shown in Fig. 6a.

In view of the inclusion of the curvature length $\lambda_2$, we analyzed the residuals of the previous statistical model and plotted them as a function of $\lambda_2$ (Fig. 6b). We find a correlation coefficient of $R^2 = 0.295$, which indicates only a small benefit of including $\lambda_2$ in the analysis. The respective statistical model

$$\frac{\mu_2}{2\mu_1} = n_0 + n_1\lambda_1 + n_2\lambda_2 \tag{22}$$

15 yields optimal parameters $n_0 = 3.95 \times 10^{-3}$ mm, $n_1 = 1.50$ and $n_2 = -2.46 \times 10^{-1}$ with a correlation coefficient $R^2 = 0.949$. The $p$-value for the intercept $n_0$ is 0.36. For $n_1$ and $n_2$ the $p$-values are again very low.

We have heuristically found a possibility of improving Eq. (22) even further. This was achieved by including a factor $(1-\phi)$ on the left-hand side. More precisely, we tried

$$\frac{(1-\phi)\mu_2}{2\mu_1} = q_0 + q_1\lambda_1 + q_2\lambda_2 \tag{23}$$

20 as a statistical model. Here the optimal parameters are $q_0 = -1.23$ mm, $q_1 = 1.03$, and $q_2 = -1.98 \times 10^{-1}$. The $p$-values for all coefficients are negligible and the correlation coefficient is $R^2 = 0.980$. The results are shown in Fig. 6c. The origin of the improvement of Eq. (23) over Eq. (22) is discussed in section 5.4.

### 4.5 Relating microwave metrics and optical metrics

In the previous sections we found a statistical relation between the correlation length and the geometrical scales $\lambda_1$ and $\lambda_2$ 25 on one hand and a relation between the exponential correlation length and the chord length moments on the other hand. Both findings can be recast into a direct connection between the moments of the chord lengths $\mu_1$ and $\mu_2$ and the exponential correlation length $\xi$. We express this relation in the statistical model

$$\xi = c_0 + c_1(1-\phi)\mu_1 + c_2 \frac{(1-\phi)\mu_2}{2\mu_1}. \tag{24}$$

We obtained the correlation coefficient $R^2 = 0.985$ for the optimal parameters $c_0 = 9.28 \times 10^{-3}$ mm, $c_1 = -7.53 \times 10^{-1}$, 30 $c_2 = 2.00$. This final relation Eq. (24) significantly improves both models Eq. (18) and Eq. (19).




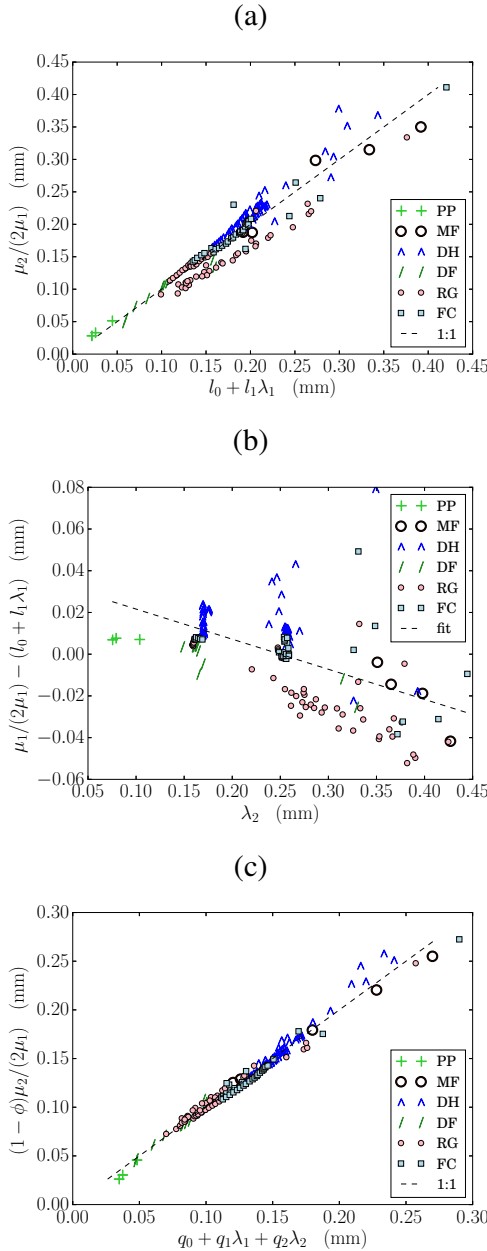

**Figure 6.** Scatter plots of a) the statistical model see Eq. (21) predicting $\mu_2/2\mu_1$ depending on the optical diameter $\lambda_1$, b) the residuals of $\mu_2/2\mu_1$ and the statistical model Eq. (21) versus the curvature length scale parameter $\lambda_2$, c) the statistical model predicting $(1-\phi)\mu_2/2\mu_1$ (see Eq. (23)) depending on the optical diameter $\lambda_1$ and the curvature length $\lambda_2$.





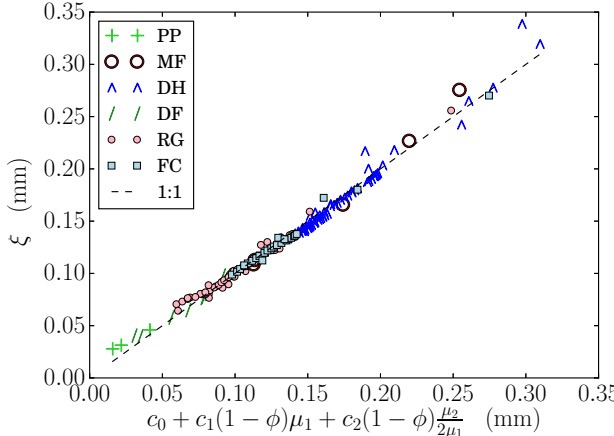

**Figure 7.** Scatterplot of the exponential correlation length $\xi$ versus the statistical model Eq. (24) that depends on the first and second moment of the chord length distribution, $\mu_1$ and $\mu_2$.

## 5 Discussion

### 5.1 Retrieval of size metrics from $\mu$CT data

Retrieving geometrical properties of the ice-air interface from tomography data must be generally taken with care. In addition to the uncertainties related to filtering and segmentation pointed out by Hagenmuller et al. (2016), the present analysis and

5 cross-validation of a curvature metric imposes additional requirements on the smoothness of the interface. The subtle influence of the smoothing parameter on the surface area $s$ and averaged mean and Gaussian curvatures $\overline{H}$ and $\overline{K}$ is apparent from Fig. 2. Naturally, $\overline{H^2}$ is most sensitive to smoothing. We found a competing performance of $\lambda_1$ and $\lambda_2$ with the smoothing parameter when comparing the triangulation based estimates with the correlation function based values. The agreement for the surface area seems to be best with smoothing parameter $S = 50$. In contrast, more smoothing is indeed required to obtain an agreement

for the curvature length. This higher sensitivity on the smoothing parameter is reasonable, since curvatures are defined by surface gradients which are more sensitive to a smooth representation than the surface area. The competing behavior is caused by the smoothing filter, which neither preserves volume nor the surface area of the enclosed ice upon smoothing iterations. This causes the drop in agreement for $\lambda_1$ in Fig. 2 (left, middle) with increased smoothing. As a remedy, more sophisticated smoothing filters could be used which, for example, ensure the conservation of the enclosed volume (Kuprat et al., 2001). Such

problems could be partly avoided by computing normal vector fields and curvatures directly from voxel-based distance maps (Flin et al., 2005). A detailed comparison of all these different methods however, is beyond the scope of this paper.



## 5.2 Linking exponential correlation lengths and curvatures

Accepting the methodological uncertainties discussed in the previous section, we shall now discuss our findings of the statistical analysis and their relevance for the interpretation of snow microstructure.

As a first step we have analyzed the statistical relation between correlation length and grain size (Mätzler, 2002) which is consistent with our data. Compared to Mäzler's model that predicts $a_1 = 0.75$, we find a slightly higher value of $a_1 = 0.79$. This can be explained by a large number of depth hoar samples where $\xi$ is generally higher than for other snow types. This is also suggested by the data from Mätzler (2002, Tab. 1), which indicates an influence of snow type or grain shape. This influence was made quantitative by the subsequent analysis where we found a clear improvement of the prediction of the exponential correlation length when incorporating the curvatures length as an additional size metric (Fig. 4c). The quantitative improvement on the statistical model Eq. (17) by using Eq. (19) or Eq. (24) is given by the increase in the correlation coefficient from $R^2 = 0.733$ to $R^2 = 0.922$ and $R^2 = 0.985$, respectively. To ensure that the inclusion of an additional parameter in Eq. (19) and Eq. (24) indeed improves on eq. (17), we have employed the Akaike information criterion (AIC) measure (Akaike, 1998). This allows us to discern if the improvement of the correlation coefficient is trivially caused by an increasing number of fit parameters or an actual improvement on the likelihood of the fit due to the relevance of the added parameters. Absolute AIC-measures have no direct meaning, however a decrease of at least $2k$ between two models, where $k$ is the number of extra parameters, implies a statistical improvement. For our case $k = 1$ the difference in the AIC-measure between Eq. (18) and Eq. (19) is 177 and the AIC difference between models Eq. (19) and Eq. (24) was 275, which confirms the statistical significance of the model Eq. (24).

All statistical models indicate that at least two different length scales $\lambda_1$ and $\lambda_2$ or $\mu_1$ and $\mu_2$ are required to obtain a reasonable prediction of the exponential correlation length. While $\lambda_1$ and $\mu_1$ are both trivially related to the optical radius via Eq. (1) and Eq. (13), the two other size metrics $\mu_2$ or $\lambda_2$ significantly increase the performance of the statistical model. As further detailed below, both parameters can be regarded as a two possibilities of defining grain shape.

## 5.3 The notion of grain shape

The international classification for seasonal snow on the ground (Fierz et al., 2009) considers grain shape as the morphological classification into snow types. This is motivated by the common but loose perception of shape as the basic geometrical form of constituent particles. It is clear that grain shape remains a vague concept unless it is formulated in terms of quantities which are unambiguously defined on the 3D microstructure.

Local curvatures are often regarded as shape parameters and used to characterize snow on a more fundamental level. The relevance of the mean curvature is described and analyzed in detail in Calonne et al. (2015), where morphological transitions (e.g, faceting) of snow during temperature gradient metamorphism are visible in the distribution of mean curvatures. The present description of grain shape in snowpack models (Lehning et al., 2002; Vionnet et al., 2012) is in fact based on the variance of the mean curvature, by the "sphericity" parameter as introduced by Brun et al. (1992). There are attempts to measure the sphericity from digital photographs as described by Lesaffre et al. (1998) and Bartlett et al. (2008). This definition is valid





only in two dimensions and therefore difficult to compare to their 3D counterparts in Calonne et al. (2015). Another aspect of shape is captured by the averaged Gaussian curvature $\overline{K}$. Though $\overline{K}$ is computed from local properties of the interface, it has a strict topological meaning due to its relation to the Euler characteristic $\chi$ via Eq. (9). The Euler characteristic was e.g. used by Schleef et al. (2014) to characterize microstructural changes during densification. As a topological quantity, $\chi$ is by definition

strictly *independent* of local (shape) variations of the ice-air interface. We found however, that the contribution $\overline{K}/3$ in $\lambda_2$ from Eq. (8) ranges from 1-13% and is on average 3.7 % of $\overline{H^2}$. Hence the curvature length $\lambda_2$ is dominated by the second moment $\overline{H^2}$, and thus closely related to the variance of the distribution of mean curvatures, which is a well-defined shape concept for the 3D microstructure.

     There is a conceptual pitfall associated with shape metrics of 3D microstructures. To illustrate this, we consider a microstruc-

ture of polydisperse spherical particles. The definition of grain shape from the classification (Fierz et al., 2009) would assign a spherical shape to this microstructure, while the averaged squared mean curvature $\overline{H^2}$ would be rather governed by the variance of particle radii. This indicates that polydispersity must also be considered as a particular aspect of shape. The equivalence between polydispersity and shape can be made more rigorous as pointed out by Tomita (1986): a low-density assembly of irregularly shaped but identical particles can always be mapped uniquely, by solving an integral equation, onto a system of

polydisperse spherical particles *if* only the correlation function is considered. Irregularity can be equivalent to polydispersity. Hence, snow types which can be clearly discerned visually might still have very similar physical properties. Shape must be generally understood as a distribution of size metrics. This also explains why the objectively defined shape parameter $\lambda_2$ cannot be mapped directly onto the classical definition of grain type from Fierz et al. (2009).

### 5.4   Linking optical and microwave metrics

Finally, we turn to the implications of different descriptions of grain shape for modeling microwave scattering or geometrical optics in snow.

     The exponential correlation length must be understood as a proxy to characterize the entire correlation function by a single length scale. By construction, this single length scale contains signatures of both, properties that dominate the behavior of the correlation function for small arguments ($\lambda_1$ and $\lambda_2$) and other properties that dominate the tail-behavior of the correlation

function for large arguments. Within the scope of such a single length scale metric, we found clear evidence from the statistical relation Eq. (19) that the tail is already largely determined by properties of the correlation function at the origin ($\lambda_1$ and $\lambda_2$). This seems surprising at first sight. Why should local aspects of the interface ($\lambda_1$ and $\lambda_2$) determine the (non-local) decay of structural correlations ($\xi$) relevant for microwave scattering? To illustrate our explanation for this finding, we resort to a particle picture and consider a dense, random packing of monodisperse hard spheres. For such a packing, the particle "shape"

is trivial and fully determined by the sphere diameter $d$, which determines the slope of the correlation function at the origin. However, also particle positions and thus the decay of correlations is fixed by $d$. This becomes obvious from the representation $C(r) = n v_{\mathrm{int}}(r) + n^2 v_{\mathrm{int}}(r) * h(r)$ for the correlation function for such as system at number density $n$ (Löwe and Picard, 2015). In this representation, the spherical intersection volume $v_{\mathrm{int}}$ and the statistics of particle positions $h(r)$ both depend on $d$. Now imagine that each sphere is deformed by a hypothetical, volume-conserving re-shape operation to an irregular,





**Table 1.** Determination of the absorption coefficient $\alpha$ (Warren and Brandt, 2008) and the fraction of the first and second order of Eq. (12) including the standard deviation $\sigma$.

| wavelength ($\mu$m) | $\alpha$ (m$^{-1}$) | $\overline{\mu_2/2\mu_1}\alpha$ (%) | $\sigma$ (%) |
|---|---|---|---|
| 0.90 | 4.1 | $7.6 \times 10^{-2}$ | $2.6 \times 10^{-2}$ |
| 1.31 | $1.2 \times 10^2$ | 2.1 | $7.2 \times 10^{-1}$ |
| 1.63 | $2.0 \times 10^3$ | 37 | 13 |
| 1.74 | $1.1 \times 10^3$ | 20 | 6.8 |
| 2.00* | $9.4 \times 10^3$ | $1.7 \times 10^2$ | 60 |
| 2.26 | $1.1 \times 10^3$ | 20 | 7 |

* wavelength is not used for optical measurements

non-convex particle, which is still located at the center of the original sphere. Due to re-shaping, the parameter $\overline{H^2}$ would increase. After the re-shape, neighboring particles would overlap (on average), since their maximum extension must have been increased compared to the sphere diameter. To recover a non-overlapping configuration, all particle positions must be dilated. The latter, however, also affects the tail of the correlation function. This is exactly what we observe: the "shape of structural units" in snow, as exemplified by $\overline{H^2}$ is always correlated with the "position of the structural units" in space. We note that such a particle analogy has clear limitations and only serves here as an attempt to illustrate the rather abstract statistical relations between different length scales. They must be taken with caution, since snow is a bicontinuous material if probed by $\mu$CT, and individual particles cannot be distinguished.

The previous analogy also helps to understand why geometrical optics of snow should be related to microwave scattering, despite the difference in wave lengths by orders of magnitude. For snow optics, it has been shown that shape influences the penetration of light (Libois et al., 2013). The authors conclude that a collection of spheres cannot sufficiently predict irradiance profiles in snow due the underestimation of the asymmetry factor $g^G$. This factor is known to include shape of different grain types as predicted by the theory from Kokhanovsky and Zege (2004). However an expression of the shape parameter $B$ in terms of the microstructure is not provided by the theory. The analysis of Malinka (2014) shows that the optical properties can be expressed in terms of the Laplace transform $\widehat{p}(\alpha)$ of the chord length distribution, which has to be evaluated at the absorption coefficient of ice, $\alpha = 2\pi\kappa/\lambda$, where $\lambda$ is the wavelength and $\kappa$ the imaginary part of the refractive index. Since for most wavelengths in the visible and infrared regime $\alpha\mu_1 \ll 1$ is small, the Laplace transform Eq. (10) can be approximated by a few terms in the expansion Eq. (12). The results in Malinka (2014) are mainly based on the Laplace transform of an exponential, $\widehat{p}(\alpha) = 1/(1 + \mu_1\alpha)$, which only involves $\mu_1$ (or the optical radius via Eq. 1). Assessing typical values for $\alpha$ allows us to estimate the relative importance $\alpha\mu_2/2\mu_1$ of the second-order term compared to the first-order term in the expansion Eq. (12). Typical values for $\alpha$ are obtained by using the values for $\kappa$ provided by Warren and Brandt (2008). The ratio $\alpha\mu_2/2\mu_1$ is calculated for typical wavelenghts and shown in Table 1. Wavelengths are selected to match common optical methods, namely 0.9 $\mu$m (Matzl and Schneebeli, 2006), 1.31 $\mu$m (Arnaud et al., 2011), and the SWIR wavelengths 1.63 $\mu$m, 1.74 $\mu$m and





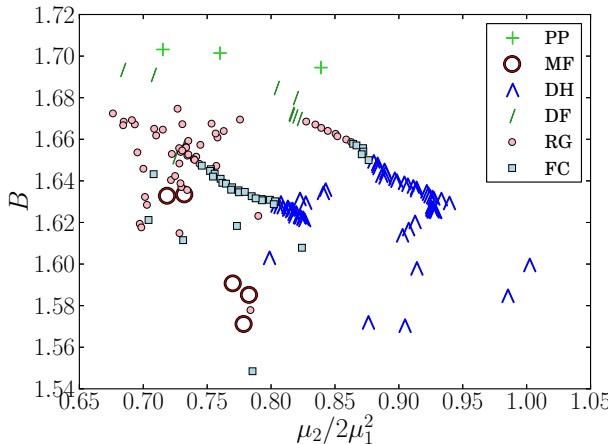

**Figure 8.** Scatterplot of the dimensionless quantity $\mu_2/2\mu_1^2$ and the optical shape factor $B$ evaluated for refractive index at wavelength $\lambda = 1.3\,\mu$m.

2.26 $\mu$m used by Domine et al. (2006). We added the wavelength 2.00 $\mu$m, which is not used by any instrument, but has the highest value for $\alpha$ in this range. Note that the standard deviation $\sigma$ is high as a result of the variations due to grain shape. The lowest values of $\mu_2/2\mu_1$ are found for fresh snow (PP) and highest for depth hoar (DH) and melt forms (MF). Given the order of magnitude, it seems likely that shape corrections could be measured by some SWIR based optical techniques. To

confirm the relevance of the shape correction from a different perspective, we can directly compute the optical shape parameter $B$ in terms of $\mu_1, \mu_2$. It is straightforward to derive an expression for $B$ using (Libois et al., 2013; Malinka, 2014) as shown in the Appendix A. The results are shown in Fig. 8 where $B$ is shown as a function of the dimensionless quantity $\mu_2/2\mu_1^2$ which can be constructed from the two relevant parameters. The range of values $B \in [1.54, 1.72]$ is well within the range $B \in [1.25, 2.09]$ obtained by ray-tracing calculations for different geometrical shapes (Libois et al., 2013). Further details

remain to be elucidated by combining tomography imaging together with optical measurements or pore scale simulations. Along these lines our results suggest a new route of assessing the remaining discrepancies in Haussener et al. (2012) using the moments of the chord length distribution.

The established connection between $\mu_2$ and shape (via $\lambda_2$) is demonstrated by the statistical model Eq. (23) and the residual analysis (Fig. 6). Together with the relation between $\xi$ and $\lambda_2$ discussed in 5.2, we have finally established a connection between

all involves size metrics. This leads to the statistical relation Eq. (24), which involves density, the microwave metric $\xi$ and the optical metrics $\mu_2$ and $\mu_1$.

The statistical relations between all the size metrics was motivated by the connection between chord length distributions and correlation functions. This connection is an old topic which was raised in the context of small angle scattering half a century ago (Méring and Tchoubar, 1968). The approximation Eq. (14) used here actually contains two different approximation steps. A

first simplification comes from the assumption that consecutive ice chords are statistically independent. Such an approximation




was used by Roberts and Torquato (1999) to derive an exact, but more complicated, relation between the Laplace transforms of the ice chord length distribution and the correlation function. A similar result was obtained by Levitz and Tchoubar (1992). The used relation Eq. (14) underlies even an additional approximation of strong dilution of the inclusion particles (Méring and Tchoubar, 1968). Despite the two-step approximation outlined above, we however confirmed that Eq. (14) has a practical value

and yields three, qualitatively consistent results for different snow types (Fig. 5). First, it captures the considerable variations of the position of the maximum, the width, and decay of the chord length density function. Second, it leads to the suggested Eq. (22) which indicates that moments of the chord length distribution and the second derivative of the correlation function must be related. An heuristically found improvement on Eq. (22) by including the term $(1 - \phi)$ in Eq. (23) is not surprising since snow is not a dilute particle system and corrections containing $\phi$-terms must be expected. Third, the relation Eq. (14)

predicts that the chord length distribution tends to zero for small values i.e. $p(\ell = 0) = 0$ (as confirmed in Fig. 5). This is a direct consequence of a smooth interface as shown in Wu and Schmidt (1971). The latter work also derived the real space expansion of the chord length distribution which can be written as $p(\ell) = 6\ell/\lambda_2^2 + \mathcal{O}(\ell^3)$. This result based on the assumption of a dilute suspension of identical, randomly oriented particles, can be taken as an independent confirmation that the variance of the chord length distribution $\mu_2 - \mu_1^2$ must be related to the interfacial curvatures via $\lambda_2$. Under the minimal assumption that

the chord length distribution is governed by at least two independent length scales, the width of the distribution must result from a competition of the rate at which the probability increases for small arguments $\ell$ (equal to $6/\lambda_2^2$) and the rate at which probability density decays to zero for large arguments $\ell$ (which must contain the optical radius $\lambda_1$).

An obvious drawback of Eq. (14) is, however, also revealed by Fig. 5 for the RG snow. Due to the quasi-oscillations in the correlation function (cf. (Löwe et al., 2011)), $A(\ell)$ and its second derivative assume negative values, which would imply

negative values for $p(r)$ via Eq. (14). This is in contradiction to the meaning of $p(r)$ as a probability density. The results from Roberts and Torquato (1999) for similar systems of oscillatory correlation functions indicate that the more sophisticated approach using numerical Laplace inversion seems to be a remedy, however this is beyond the scope of the present work.

As a convenient side product of our analysis, we obtained an approximate relation for the second moment of $\mu_2$ of the chord length distribution in terms of the curvature length $\lambda_2$ (predominately via $\overline{H^2}$). The parameter $\overline{H^2}$ has also been used for shape

recognition in stereology for a long time and can be obtained from particular vertex and edges counting algorithms, as shown by Cahn (1967) and DeHoff et al. (2015). An analytical relation between the chord length distributions and curvatures was, however, never derived. Due to the lack of closed form expression for $\mu_2$, our results may be relevant also for other applications.

## 6   Conclusions

In this work we have we analyzed snow microstructure and suggested a size metric which objectively, but not uniquely,

characterizes shape from the expansion of the correlation function in terms of interfacial curvatures. We have shown that the geometrical interpretation of the shape parameter is indeed correct by a comparison to VTK-based triangulation methods. This also highlighted the remaining difficulties when processing the ice-air interface, such as smoothing. Independent of these difficulties, the shape analysis allowed us to improve a widely used statistical model for the exponential correlation length





(as a key size metric for MEMLS based microwave modeling) from the optical radius by including shape via curvatures. Alternatively, the exponential correlation length can also be expressed in terms of moments of the chord length distribution (as the key metric for geometrical optics modeling). We analyzed the connection between chord length distributions and correlation functions which was suggested by old arguments from small angle scattering. Loosely speaking, the established connection

states that local shape of irregular snow grains (determining optical response via the chord lengths or curvatures) and the packing of these irregular grains (determining microwave response via the correlation length) is intimately correlated. Our results suggest a new experimental route to connect optical in-situ field measurements with microwave measurements. This requires to design an experimental method which is able to retrieve the $\mu_2$ corrections (shape) in the optical properties when compared to the $\mu_1$ term (optical radius). This seems possible given the predicted values for the optical shape factor $B$. In a

second step, using the statistical relation Eq. (24), a direct connection to the correlation length can be made. Even by treating snow here as an isotropic medium (by averaging all quantities over directions) we have found statistically robust relations between all size metrics. With ongoing progress in models for the correlation function that include anisotropy and more general forms other than exponential ones, we can expect further refinement in the relation between optical and microwave metrics in the future.

**Appendix A:  Optical shape factor $B$ from moments of the chord length distribution**

To derive an expression of the optical shape factor $B$ in terms of the moments of the chord length distribution, we start from expression (Libois et al., 2013, Eq. 6) for the single scattering co-albedo $(1 - \omega)$ as defining equation

$$(1 - \omega) = B \frac{\gamma V}{2\Sigma}, \tag{A1}$$

which relates $B$ to the average volume of a particle $V$, the average projected area of a particle $\Sigma$, and the absorption coefficient

$\gamma$. This can be reformulated in the chord-length picture by using (Malinka, 2014, Eq. 6). Then, adopting the notation of the present paper, the relation (A1) can be written as

$$(1 - \omega) = B \frac{\alpha \mu_1}{2} \tag{A2}$$

Using the expression of the single scattering albedo from Malinka (2014, Eq. 56), inserting (Malinka, 2014, Eq. 29,42,49,18) and re-arranging terms we obtain

$$(1 - \omega) = \frac{T_{\text{out}}(n)}{1 + \frac{T_{\text{out}}(n)}{n^2} \frac{\widehat{p}(\alpha)}{1 - \widehat{p}(\alpha)}} \tag{A3}$$

in terms of the real part of the refractive index $n$, the averaged Fresnel transmittance coefficient $T_{\text{out}}(n)$ (given by Malinka (2014, Eq. 19) in closed form) and the Laplace transform of the chord length distribution $\widehat{p}(\alpha)$. By comparing (A2) and (A3), and taking into account an additional factor of 2 between (Malinka, 2014) and (Libois et al., 2013) due to a different treatment





of the extinction efficiency, we end up with

$$B = \frac{1}{\alpha\mu_1} \frac{T_{\text{out}}(n)}{1 + \dfrac{T_{\text{out}}(n)}{n^2} \dfrac{\widehat{p}(\alpha)}{1 - \widehat{p}(\alpha)}} \tag{A4}$$

Complemented by the approximation (12) for the Laplace transform $\widehat{p}$, the expression (Malinka, 2014, Eq. 19) for $T_{\text{out}}(n)$ this yields an expression of the shape factor $B$ in terms of the first and second moment, $\mu_1, \mu_2$, of the chord length distribution, the real part of the refractive index $n$ and the absorption coefficient $\alpha$.

*Acknowledgements.* The authors thank G. Picard for a constructive feedback on an earlier version of the manuscript and S. Torquato for helpful clarifications on the factor 2/3 between $s_{\text{mf}}$ and $s_{\text{cf}}$. M. Lehning provided valuable suggestions on the statistical methods. The work was funded my the Swiss National Science Foundation via Grant No. 200021_143839.



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
