# Peer review of "Relating optical and microwave grain metrics of snow: The relevance of grain shape"

_The Cryosphere, 2016_

## Referee Comment (RC1) · A. Malinka (Referee) · 12 Jun 2016

The article presents a study, important for optics and physics of snow. It improves our understanding of snow microstructure. The authors attract our attention to the importance of the third term of the expansion of the correlation function, related to the curvature of the air-ice interface. One of the achievements of the work is the correlations between the microstructure parameters, both short-scale and long-scale, which are established experimentally by investigating the snow samples.

There are some points to discuss.

1. The authors state that the second term in $A(r)$ expansion (and therefore $p(0)$) is equal to zero and explain that this is a direct consequence of the interface smoothness. However, the widely used (e.g., by Debye) exponential function for $A(r)$ has the obviously nonzero second term. At the same time, there are interface models, such as the Switzer model, that provides strictly exponential correlation function. Particularly, in the Switzer model the space is dissected by a set of random planes into random polyhedrons and the resulting polyhedrons are assigned to ice with the probability $\phi$ and to air with the probability $1-\phi$. This interface is not smooth: it has plane facets and sharp edges. Obviously, it doesn't match the morphology of aged snow, but fresh snow seems to be much closer to the Switzer interface than to smooth one, because of the facets and edges of ice crystals. With this in view and taking into account the importance of the exponential correlation function, it would be extremely desirable to discuss the facet-edge interface and its relationships with the smooth one.

2. The motivation of Eq. (15) looks invalid. In general, the integral of a function from 0 to $\infty$ is not determined by its behavior at 0. More precisely, the authors say that "A(r) depends at least on two independent length scales, λ1 and λ2" and further "In the absence of other relevant scales…" But 'at least' doesn't mean 'only'. It is obvious that, as λ1 and λ2 are the coefficients of expansion at 0, there are other terms and, hence, other independent length scales at the interval $(0, \infty)$. Figure 1b clearly demonstrates the idea that the integral is not determined by the behavior at 0, because the contribution of the function tail can be of any value.

   This note doesn't affect the further results of the work, because the authors show that short-length and tail scales must correlate and try to explain why. However, at the stage of Eq. (15) this statement looks ill-founded.

   Let me suggest the idea. As the value of the correlation length $\xi$ is derived from the fitting the correlation function by the exponential at the whole interval, the estimation

$$\int_0^\infty A(r)dr = \xi$$

   looks much more reliable.

   Partially, this implication is confirmed by the fact that, when considering the correlation between $\xi$, $\mu_1$, and $\mu_2$, the obtained correlation coefficient at $\mu_2$ is higher than that at $\mu_1$.

   (Minor: the differential $dr$ is missing in the integral).

3. Page 14, line 25: "In the previous sections we found a statistical relation <…> between the exponential correlation length and the chord length moments on the other hand." I guess the authors wanted to say "between the geometrical scales λ1 and λ2 and the chord length moments," because the relation between the exponential correlation length and the chord length moments is considered just below.

4. Introducing the factor $1-\phi$ into Eq. (24) the authors go back to the length λ1 in the second term by virtue of Eq. (13). This is worth to note. Also, with the factor $1-\phi$ in Eq. (23) the second term turns to $\mu_1$. In the whole, it is worth to underline that $\lambda_1$ and $\mu_1$ are always related with

Eq. (13) and indeed $\mu_1$ have the meaning of the optical size, being exactly $\mu_1 = \dfrac{2}{3} d_{opt}$ independently of the snow density.

5. Page 19, line 18-19. "The results in Malinka (2014) are mainly based on the Laplace transform of an exponential, p($\alpha$) = 1/(1+ $\mu$1$\alpha$), which only involves $\mu$1 (or the optical radius via Eq. 1)." This is not completely true, because the exponential law is considered only as an example, though very important one. I would just delete this sentence, because it doesn't carry important information.

6. Page 19, line 20, table 1: "relative importance $\alpha\mu2/2\mu1$ of the second-order term compared to the first-order term in the expansion Eq. (12)." This value doesn't look very informative. I think that much more informative will be the value, proportional to the variance $\alpha(\mu_2 - \mu_1^2)/2\mu_1$, because it will give the deviation from the exponential law.

7. It would be nice to consider these relations taking into account the relationship between $A(r)$ and $p(l)$ in the general case of a dense medium, not restricted by the dilute one.

**Reference**
P. Switzer, "A random set process in the plane with a Markovian property," *Ann. Math. Statist* **36**, 1859-1863 (1965).

---

## Referee Comment (RC2) · Q. Libois (Referee) · 6 Jul 2016

**General comments**

This paper addresses the relation between the grain metrics commonly used to model snow optical and microwave properties. At first order, snow microwave properties are governed by the exponential correlation length ξ while snow optical properties firstly depend on snow specific surface area (SSA). However, at second order snow grain shape also affects snow radiative properties. From this statement, statistical relations are derived that make the link between snow microstructure characteristics (curvatures) and snow physical properties. The relation between ξ and SSA is thus improved compared to previous empirical relations, by adding a contribution of snow grain shape. The general theoretical framework of Malinka (2014) is then used to show that snow optical properties depend on the moments of the chord length distribution. Based on this framework, another statistical relation is derived to express the second moment of the chord length distribution in terms of microstructure length scales. From this, a statistical relation between ξ and the first two moments of the chord distribution is derived. This suggests that shape parameters derived from optical measurements could be used as inputs for snow microwave modeling. This point is supported by comparing the values of the optical shape parameter *B* deduced from Malinka (2014) theory to values determined experimentally.

The paper is overall well written and pleasant to read, the objectives are well defined at the end of the introduction. The theoretical background is nicely presented and clearly outlines the problem. The approach is original and takes advantage of recent works in snow optics. It also applies the statistical properties of general random heterogeneous materials to the case of snow, thus linking rather theoretical studies and practical cases as illustrated by the use of μCT images of snow samples. The authors stress the need for a unified definition of grain shape and propose mean curvatures as such definition. They show that both microwave and optical properties can be expressed in terms of SSA and mean curvatures. Their approach is supported by the analysis of a large set of μCT images. They also provide valuable physical insight on the representation of snow microstructure as a particulate or heterogeneous medium. For these reasons, I recommend this paper be published in The Cryosphere. However, a number of critical points should be addressed before publication, related in particular to the fundamental assumptions underlying the presented theoretical framework.

**Specific comments**

The theoretical framework presented in this paper strongly relies on critical assumptions that are not sufficiently discussed, although several important results largely depend on them.

1) Throughout the text, snow is considered isotropic and the derivations significantly rely on this critical assumption. Although this assumption is clearly stated, several details are lacking to convince the reader that the results remain reliable. First, more details on the investigated snow samples should be provided. So far only 3 lines (section 3.1) present these critical elements of the study, which is not enough. Do these samples consist of sifted snow, natural snow samples taken in the field without perturbing the microstructure, snow samples resulting from metamorphism experiences in the laboratory...? It is clear that depending on the origin of the samples, the isotropic hypothesis is more or less acceptable. For instance depth hoar is known to be highly anisotropic and can hardly be investigated under this hypothesis. The authors should consider removing highly anisotropic snow samples if they do not fit in the theoretical background.

At the same time, the authors do have the necessary material to further discuss the isotropic hypothesis because the parameters are obtained from averages over the 3 directions x, y and z. Giving a hint of the actual anisotropy from the analysis of these 1-D parameters might help the interpretation of the data and estimate the associated uncertainties.

2) In this study, the successive chords in snow are assumed independent, which is a strong assumption not really defended by the authors. This same assumption was used by Malinka (2014) who considered a random medium, whose optical properties where then derived. However, this author clearly states in his conclusion that: "*The requirement of stochasticity is mandatory: the facets orientation and the ray path length inside solid or voids must be independent variables. [...] The question of applicability of the model to any particular medium should be considered separately based on compliance with the experimental data.*" Practically, one might expect light rays to be trapped in snow grains or selectively focused in preferential location, which would result in different chords having different realization probabilities.

A critical consequence of the random distribution hypothesis is that at low ice absorption, the optical properties of this random medium do not depend on the shape parameter (see e.g. eq. (25) of Malinka (2014) from which $B$ can easily be derived). This is somehow contradictory with the definition of grain shape, which is expected to impact snow optical properties in the standard particular representation.

An alternative approach could be to validate this random medium assumption by comparing the values of $B$ retrieved from Malinka (2014) to those determined by Libois et al. (2014), which are very similar. Once the random medium hypothesis is somehow validated, then the shape parameter only impact optical properties at more absorbing wavelengths. An important corollary of this would be that only optical measurements at relatively absorbing wavelengths would contain information about snow grain shape.

3) When it comes to the analysis of μCT images, the question of voxel size (ie resolution) is not enough discussed. In fact, the resolution varies from a set of measurements to another and is generally not that small compared to snow size metrics. This probably has an effect on the derived results and might explain why different subsets of points appear on several Figures (e.g. 2 bottom left and 4a). The smoothing parameter is discussed in sufficient details but resolution is probably an issue as critical.

4) Although snow optical properties equally depend on the parameters $B$ and $g$, the paper is mostly focused on $B$. The analysis presented for $B$ can very easily be extended to $g$. This would be more exhaustive because all parameters relevant to snow optics would be tackled, as all parameters (actually only ξ) needed for snow microwave modeling are.

5) The manuscript would benefit from a slight reorganisation of some parts because redundancy is found at several points and excessive details sometimes pollute the paper. Some elements are given too early (e.g. details about the Euler characteristic that should probably not be mentioned before the discussion section), some others should be provided in a different order (more details are provided along the technical comments). Also sections 3.3 and 3.4 could probably be merged.

6) The authors make their best to infer the shape of the statistical relations from theoretical backgrounds. However, this often adds noise to the paper because 1) the underlying assumptions are often very restrictive and not applicable to snow (dilute medium, random medium, use of Taylor expansion at 0 for estimating functions at infinity...) and 2) these statistical relations are eventually

revisited by adding terms. I think there is no problem assuming a certain type of relation, and then testing it with the available data. For sure, the type of relation can be suggested by a rapid analysis of existing formulae, but there is no need trying to justify it too much. In this context, I would suggest to remove the unnecessary calculations and reformulate the section around Eqs. (14) and (15). For instance the authors could say that they show the validity of Eq. (14) from images, even though initially this relation is only valid to restricted cases. All the attemps to justify this equation are unncessary.

7) The authors should give a consistent name to all important quantities $\xi$, $\lambda_1$, $\lambda_2$, $\mu_1$, $\mu_2$ and keep those names all along the manuscript. For instance, exponential correlation length and correlation length are sometimes used alternatively without a clear distinction. Porod length, optical diameter and curvature length are used sporadically as well.

8) At the light of the comments above, it will probably be necessary to rewrite the last section of the discussion (5.4).

**Technical comments**

Title

Could "snow grain size" be used instead of "grain metrics of snow"?
Alternative suggestions (these are only suggestions):
- "Relating optical and microwave snow grain size: The importance/relevance of using/considering grain shape"
- "Accounting for snow grain shape to improve the relation between optical and microwave snow grain size"

Abstract

p.1 l.1: rephrase to better compare the roles SSA and exponential correlation length play in determining snow optical and microwave properties.
Either from the physical point of view: "microwave emissivity/properties mostly depend(s) on the exponential correlation length".
Or from the modeling point of view : "the exponential correlation length is the relevant quantity in most snow microwave models" or "the exponential correlation length is used to simulate snow microwave properties"

p.1 l.3: a microwave model is not "forced" by optical measurements, it uses quantities derived from optical measurements (e.g. SSA) as inputs. Forcing more generally refers to something external to the system (e.g. boundary conditions). This is correctly said p.2 l.9.

p.1. l.3: "the understanding of $\xi$" is vague. Simply say "To refine this relation between...]"

p.1 l.5: it is a statistical *relation* more than a *prediction*

p.1 l.8-9 : maybe remove this sentence because it does not provide additional information about the results. Also, it is somehow questionable in terms of applicability within the present theoretical framework. Keep it for the body of the manuscript.

p.1 l.10 : *B* is called the absorption enhancement parameter. Consider doing the same calculations with *g*.

p1. l.10 : the last sentence of the abstract is not clear. Maybe say "Our results suggest that optically derived shape parameters can be used to refine the estimation of ξ".

Introduction

p.1 l.16-19 : maybe invert the order of the two sentences to keep chronological order

p.2 l.4 : "with the MEMLS model" instead of "is used"

p.2 l.14 : "though less significant..." is risky because the impact can actually be significant (errors up to 50%) for BRDF or light penetration simulations for instance.

p.2 l.16 : reference to Picard et al. (2009) might be relevant

p.2 l.17 : in this study the absorption enhancement parameter *B* and asymmetry factor *g* (name these factors) are equally important, except that only *B* can be estimated from optical measurements. Note that Libois et al. (2014) experimentally determined the parameter *B* for a variety of natural snow samples.

p.3 l.1 why "systematically?"

p.3 l.12 : not clear what "images" you're talking about

p.3 l.15-17 : maybe keep those last 2 sentences for the discussion and mention it more shortly at this stage because this is hard to understand without the whole paper in mind.

Theoretical background

p.3 l.21-22 : very redundant with p.1 l. 20-21.

p.4 l.5 : why "in contrast"? Is the exponential approximation only valid for large *r* values?

p.4 l.14: use $m^2\ kg^{-1}$ instead

p.4 l.24-28 : consider mentioning the topological dimension of the mean Gaussian curvature only in the discussion, because at this stage the reader does not understand the point.

p.4 l.26: the mathematical notation is not clear. Maybe use dS or dA to explicitly state that this is an average on the surfaces? This integration element could also be moved after the integrand.

p.4 l.27: *that* the local. Why is local in parenthesis?

p.6 l.10: detail why *z* is actually small and mention in which conditions this theoretical framework is valid. This in in fact detailed below, but inverting the order might be helpful.

p.6 l.13 : to *the* theory of

p.6 l.14 : it's $4\pi$ rather than $2\pi$.

p.6 l.20: state here that the following sections investigate this issue and try to find a geometrical meaning of this second moment.

p.7 l.2 : would it be useful to briefly define the surface-void correlation function? Otherwise

p.7 l.4 : please clarify the meaning of "this is not a practical limitation"

p.7 l.1-7: since eventually the relation of Roberts and Torquato (1999) is not used, this part adds noise to the paper. Consider removing it (or mention it more concisely) if indeed it is not used.

p.7 l.12: not clear why you keep going while snow is clearly not a dilute medium. If the relation actually holds for snow (which seems to be the case as you show its consistency), state there that you demonstrate its validity for snow.

p.7 l.15: it seems that integrating by parts result in a factor $\left[l\dfrac{dA(l)}{dl}\right]_0^\infty$ . Why is it equal to 0? True for the exponential case. Idem for p.7 l.18

p.7 l.20 : the expansion is only valid for small $r$ values, while here the integration goes much beyond.

p.7 l.20-24 : This paragraph somehow adds noise to the flow of the paper. Would it be problematic to make it shorter and simply state that in Eq. (15) the integral is a function of $\lambda_1$ and $\lambda_2$ and must be of "length" dimension? I think this would not change the use of this equation later on (section 4.4). This approach would also allow the use of a constant term in the fit of Eq. (21) without further justification.

Methods

p.8 l.4 : More details about the preparation of the samples should be provided, and the isotropy of the prepared samples should be discussed. If for instance some samples obviously do not follow the isotropy requirement (e.g. depth hoar) they should be removed from the analysis.

p. 8 l.10 : the point regarding voxel size is very critical because the length scales are similar to voxel size, implying potential impact of voxelisation on the results. Can images at 18 and 50 µm be compared? See specific comment 3.

p.8 l.11 : before averaging, an evaluation of the anisotropy (or isotropy) should be given, because the whole theoretical framework is based on the isotropic hypothesis.

p.8 l.15 : Figure 1b does not really illustrate the exponential regression

p.8 l.23 : the meaning of "in view of shape" is not clear.

p.8 l23-25 : state more clearly that the section aims at validating the Eqs (6) and (8) by computing the interfacial area and interfacial curvatures.

p.8 l.30 : could this smoothing parameter be slightly more detailed, because it seems critical in the

following section. What's the typical range, what values were used in the past? For what kind of applications?

p.9 l.4 : for S = 200, the interfacial area is larger, but the points seem also more spread, which is not discussed.

p.9 l.6-11 : what is the objective of this section? Does it serve the paper? Should it be used to support the isotropic hypothesis?

p.9 l. 16 : one should be with superscript "cf"

Figure 2 (bottom left) : there seems to be 2 sets of points, one consisting of RG. Could this observation help interpreting the limitation of S = 50?

Figure 4a : there seems to be 2 sets of points. Do they correspond to similar subsets of µCT images? The same 2 sets are observed in Fig. 6a
Figures 4b and c : DH is clearly an outsider here. Is it relevant to keep it in this study?

Results

p. 11 l.11 : one extra "and"
p. 11 l.11 : is it consistent to have a $R^2$ less (0.731<0.733) for the regression with an additional parameter?

p.13 l.1 : the name of $\lambda_1$ should be consistent between titles of sections 4.1 and 4.2. In section 4.1, optical diameter is not mentioned except in the title.

p.13. l.7 : I don't really understand this justification and don't think this is necessary. I would proceed the other way round instead. The figure 4b could be discussed at the end of section 4.1 with the aim of understanding the remaining residuals. This would naturally lead to the regression Eq. (19).

p.13 l.13 and 14: Eq. (14) instead of (16)

p.14 l.3 : Eq. (15) in stead of Eq. (14)

p.14 l.17 : here you try "heuristically" a regression, which is fine. This somehow contrasts with the previous regressions that were based on the derivation of equations. This could also be motivated by the form of Eq. (13) that includes the porosity factor. I think there is no problem assuming a relation, and then testing its validity with measurements. This is sometimes easier to understand than long inexact derivations.

p.14 l.12 : it is awkward to read that the benefit is small but to see the new regression, though. I would put it more positively: "The correlation coefficient ($R^2$=0.295) is small but including $\lambda_2$ in the analysis further improves the fit".

p.14 l.24-25 : this is sometimes disturbing to read "correlation length" at some point and "exponential correlation length" later on. Please remain consistent throughout the manuscript, with each quantity ($\xi$, $\lambda_1$, $\lambda_2$) having its dedicated and constant name. Consider using "exponential" for the first part of the sentence, and "correlation length scales or Porod length and curvature lengths (for instance)" for the

second part, to make the link with Eqs. (19) and (23) more obvious.

Figure 6 : remove "see". $\lambda_1$ is not the optical diameter.

Discussion

p.16 l.2 : in complement to this discussion, this might be worth giving the sensitivity of Eq. (16) to the smoothing parameter, and possibly to the voxel size as well, if this makes sense.

p.17 l.5 : remind what grain size is because $a_1$ is the coefficient for $\lambda_1$ (which is optical diameter or grain size?)

p.17 l.6 : again depth hoar could be removed from the analysis if it does not satisfy the conditions of the theoretical framework.

p.17 l.7 : this is not clear what is also shown by those data. That the coefficient is larger for depth hoar?

p.17 l.21 : Eq. (7) instead of Eq. (1)
p.17 l.32 : there *were* attempts

p.18 l.5 : why is "independent" in italic. Idem for p.18 l.15 "if"
p.18 l.5 : where does this K/3 come from? It is K/24 in Eq. (8)

p.18 l.12 : this point is interesting, but puzzling as well. Indeed, from an optical point of view, a polydispersion of spheres will have the same "shape" parameters as a monodispersion in the geometrical optics approximation (and for low ice absorption), because B and g primarily depend and the shape, not on the size. Hence polydispersion would affect curvatures, but not grain shape as defined from an optical point of view. Said differently, a polydispersion of spheres will have optical properties similar to a monodispersion with same SSA, but different microwave properties.

p.18 l.32 : for such *a* system?

p.19 l.10 : wavelengths (in a single word?)

p.19 l.12 : the mentioned paper rather suggests that *g* for spheres is larger than *g* for snow, and that *B* for spheres is smaller than *B* for snow.
p.19 l.12 : the superscript *G* for the *g* refers to "geometrical", that does not account for the diffraction contribution to scattering. This does not change the sentence but should remain consistent throughout the paper.
p.19 l.12 : it depends on shape rather than includes it

p.19 l.16 : it's $4\pi$ rather than $2\pi$. By the way this quantity was already defined p.6. Then check the values for the following text and those shown in Table 1.

Table 1: Fraction of second to first rather than first to second order. Precise that mean and standard deviation are among all samples. Write 170 rather than $1.7 \times 10^2$. The values suggest no influence of shape at 0.9µm, which is consistent with the remark p.18 l.12. Note that eq. (5) of Malinka (2014) shows that at weakly absorbing wavelengths, *B* only depends on the real part of the refractive index.

This latter point should be further discussed to explore the validity of the random medium assumption used by Malinka (2014). In fact, this framework suggests that as long as the structure is random, shape has no impact on optical properties. This is contradictory to the fact that in the particulate representation of snow, different grain shapes result in different optical properties, even at low ice absorption.

p.20 l.6 : the authors decide to emphasize the parameter $B$, but in fact eq. (60) of malinka (2014) can also be used to express $g$ in terms of $\lambda_1$ and $\lambda_2$. This should be done to complete the analysis.

p.20 l.7 : why is the parameter $B$ shown in terms of this ratio? Is there supposed to be a visual correlation in Fig. 8? Why is the regression with respect to this particular ratio?

p.20 l.9 : Libois et al. (2014) experimentally determined the parameter $B$ for a large set of snow samples and suggest $B$ equals $1.6 \pm 0.2$. This comparison completes that with Libois et al. (2013). Note again that the range obtained in Fig. 8 results from the impact of shape at 1.3µm. This range can hardly be compared to that obtained by Libois et al.(2013,2014) obtained at visible wavelengths. The absolute values can on the contrary be compared.

p.20 l.9-12 : these sentences are not clear, and reference to Haussener et al. (2012) is very fuzzy, in particular the "remaining discrepancies".

p.20 l.15 : *involved*

p.20 l.20 : this is the very critical assumption that should be further discussed

p.21 l.l.1-16 : this part shows is partly redundant with previous parts of the text. This could be shortened.

p.21 l.11 : why is this work mentioned here and not before? Could this help to establish the semi-heuristical relations displayed all along the manuscript?

p.21 l.12-14 : Why is the variance of the chord length distribution mentioned here for the first time?

p.21. l.19 : remove parenthesis in reference

Conclusions

p.21 l.29 : extra "we"
p.21 l.29 : consider adding ($\lambda_2$) after size metric

p.22 l.9 : the meaning of "when compared to" is not clear
p.22 l.9 : Maybe say : "The consistency between $B$ values derived from the chord length distribution and those determined from optical measurements suggests such an approach is indeed possible".

Appendix

p.22 l.28 : no parentheses for the references

p.23 l.8 : *by* the Swiss...

**References:**

Haussener, S., Gergely, M., Schneebeli, M., & Steinfeld, A. (2012). Determination of the macroscopic optical properties of snow based on exact morphology and direct pore-level heat transfer modeling. *Journal of Geophysical Research: Earth Surface*, *117*(F3).

Libois, Q., Picard, G., Dumont, M., Arnaud, L., Sergent, C., Pougatch, E., ... & Vial, D. (2014). Experimental determination of the absorption enhancement parameter of snow. *Journal of Glaciology*, *60*(222), 714-724.

Malinka, A. V. (2014). Light scattering in porous materials: Geometrical optics and stereological approach. *Journal of Quantitative Spectroscopy and Radiative Transfer*, *141*, 14-23.

Picard, G., Arnaud, L., Domine, F., & Fily, M. (2009). Determining snow specific surface area from near-infrared reflectance measurements: Numerical study of the influence of grain shape. *Cold Regions Science and Technology*, *56*(1), 10-17.

---

## Author Comment (AC1) · 3 Sep 2016

Dear Aleksey Malinka,

Thank you for your detailed and careful review and the your generally positive opinion about the work. We will address all your discussion points in the following, comments are copied and replies are given in blue. We also included the additional comment we received by email. Changes to the manuscript will be documented by a track-change pdf.

Kind regards,

Quirine Krol, Henning Löwe

The article presents a study, important for optics and physics of snow. It improves our understanding of snow microstructure. The authors attract our attention to the importance of the third term of the expansion of the correlation function, related to the curvature of the air-ice interface. One of the achievements of the work is the correlations between the microstructure parameters, both short-scale and long-scale, which are established experimentally by investigating the snow samples.

**There are some points to discuss.**

1) The authors state that the second term in A(r) expansion (and therefore p(0) ) is equal to zero and explain that this is a direct consequence of the interface smoothness. However, the widely used (e.g., by Debye) exponential function for A(r) has the obviously nonzero second term. At the same time, there are interface models, such as the Switzer model, that provides strictly exponential correlation function. Particularly, in the Switzer model the space is dissected by a set of random planes into random polyhedrons and the resulting polyhedrons are assigned to ice with the probability 1-φ, and to air with the probability 1-φ. This interface is not smooth: it has plane facets and sharp edges. Obviously, it doesn't match the morphology of aged snow, but fresh snow seems to be much closer to the Switzer interface than to smooth one, because of the facets and edges of ice crystals. With this in view and taking into account the importance of the exponential correlation function, it would be extremely desirable to discuss the facet-edge interface and its relationships with the smooth one

**Reply:** It is true that the second order appears theoretically if discontinuities in the structures such as edges and corners are present. Fresh snow, as we know, contains many of these features. The ability to detect this second order term and relate it to discontinuity features is however difficult due to image resolution and noise in the data. A theoretical sharp edge would be treated practically as a rounded edge, which likely shifts weight from the second to the third order term. The resolution of our snow samples is raised by the second referee (see comment 2) As discussed there, we only find a very weak bias of image resolution on the third order term. A second argument is given by the shape of the chord length distribution that tends to zero for small chords which is a direct consequence of the absence of the second order term in the correlation function by virtue of eq.(14).

**Changes to the manuscript:** In the theoretical section we have added a sentence that mentions the role of sharp edges of the fresh snow samples. We also added the discussion on image resolution in the discussion session.

2) The motivation of Eq. (15) looks invalid. In general, the integral of a function from 0 to ∞ is not determined by its behaviour at 0. More precisely, the authors say that "A(r) depends at least on two independent length scales, λ1 and λ2" and further "In the absence of other relevant scales…" But „at least" doesn't mean „only". It is obvious that, as λ1 and λ2 are the coefficients of expansion at 0, there are other terms and, hence, other independent length scales at the interval (0, ∞). Figure 1b clearly demonstrates the idea that the integral is not determined by the behaviour at 0, because the contribution of the function tail can be of any value. This note doesn't affect the further results of the work, because the

authors show that short-length and tail scales must correlate and try to explain why. However, at the stage of Eq. (15) this statement looks ill-founded. Let me suggest the idea.

As the value of the correlation length $\xi$ is derived from the fitting the correlation function by the exponential at the whole interval, the estimation

$$\int A(r)dr = \xi$$

looks much more reliable. Partially, this implication is confirmed by the fact that, when considering the correlation between $\xi$, $\mu_1$ and $\mu_2$, the obtained correlation coefficient at $\mu_2$ is higher than that at $\mu_1$. (Minor: the differential dr is missing in the integral).

**Reply:** We acknowledge the ambiguity in motivating Eq. 15 in the present form, and for that reason we abandoned this argument, as also suggested by the second referee (see his comment 6).
The proposed idea is an interesting alternative to define and measure $\xi$. For correlation functions that are strictly exponential this definition is equivalent. This is however more in the direction of the length scale required for the microwave scattering coefficient, where the relevant scale (raised to the third power) is the zero mode of the Fourier transform of the correlation function, i.e. the volume integral over the correlation function. We will however stick here to the more "traditional" definition and estimate $\xi$ by fitting the correlations function as done in (Vallese 1981, Mätzler 2002, Calonne 2015, Proksch 2015, Löwe 2011,2013,2015)

**Changes to the manuscript:** We have changed the motivation of eq.(14).

3) Page 14, line 25: "In the previous sections we found a statistical relation between the exponential correlation length and the chord length moments on the other hand." I guess the authors wanted to say "between the geometrical scales $\lambda_1$ and $\lambda_2$ and the chord length moments," because the relation between the exponential correlation length and the chord length moments is considered just below.

**Reply:** That is correct.

**Changes to the manuscript:** We have changed the sentence accordingly.

4) Introducing the factor $1-\phi$ into Eq. (24) the authors go back to the length $\lambda 1$ in the second term by virtue of Eq. (13). This is worth to note. Also, with the factor $1-\phi$ in Eq. (23) the second term turns to $\mu_1$. In the whole, it is worth to underline that $\lambda_1$ and $\mu_1$ are always related with Eq. (13) and indeed $\mu_1$ have the meaning of the optical size, being exactly $\mu_1 = 2d_{opt}/3$ independently of the snow density.

**Reply:** We agree that we should emphasize both, the $\mu_1$ and $\lambda_1$ relation and its independency of the density.

**Changes to the manuscript:** We added a sentence in the theoretical section to emphasize the $\mu_1$ and $\lambda_1$ relation and included the $(1-\varphi)$ term in the discussion.

5) Page 19, line 18-19. "The results in Malinka (2014) are mainly based on the Laplace transform of an exponential, $p(\alpha) = 1/(1+\mu_1\alpha)$, which only involves $\mu_1$ (or the optical radius via Eq. 1)." This is not completely true, because the exponential law is considered only as an example, though very important one. I would just delete this sentence, because it doesn't carry important information.

**Reply:** We agree.

**Changes to the manuscript:** Deleted the sentence.

6) Page 19, line 20, table 1: "relative importance $\alpha\mu_2/2\mu_1$ of the second-order term compared to the first-order term in the expansion Eq. (12)." This value doesn't look very informative. I think that much more informative will be the value, proportional to the variance $\alpha$ $(\mu_2 - \mu_1^2)$ $/2\mu_1$ , because it will give the deviation from the exponential law.

**Reply:** We agree that the deviation of the exponential distribution would be illustrative here. If this deviation is defined by subtracting the two Taylor series up to the second order and normalizing by the first order term, we however end up with $\alpha(\mu_2/2-\mu_1^2)$ $/2\mu_1$. Alternatively, the deviation from an exponential can be also characterized by the ratio $(\mu_2/2\mu_1^2)$, which is exactly unity for an exponential distribution. The values found here are considerably lower (this can be directly deduced from Fig. 8 of the present manuscript). Since this Figure will be replaced according to a comment from reviewer 2, the values of this ratio will be given in the Discussion. This also confirms what is already shown in Fig.5/Fig.8, namely that the chord length distribution of depth hoar is systematically closest to an exponential.

**Changes to the manuscript:** Table 1 is adjusted and the range of values for the ratio is given in the discussion section.

7) It would be nice to consider these relations taking into account the relationship between A(r) and p(l) in the general case of a dense medium, not restricted by the dilute one

**Reply:** We actually mentioned this point explicitly in the discussion. The work (Roberts and Torquato 1999) investigated this connection for Gaussian random fields, with good agreement over a broad range of volume fractions. This also indicates that the assumption of independence of successive chords (which underlies (Roberts and Torquato 1999) does not seem to be very restrictive. Their method however requires numerical Laplace inversion and the computation of another correlation function. For Gaussian random fields the latter is known analytically, but here it would require a considerable additional effort to introduce the relevant concepts and carry out the numerics, with almost no benefit for the established connections between the length scales.

**Changes to the manuscript:** The discussion has been rewritten and this point is made clearer now.

8) The point that was raised in the email discussion: you compare the expression A2 used by Libois et al., 2013 with the expression A3 from my paper (or eq. 23 in that numbering). But expression A2 (A1) is written for small absorption only, while eq. A3 is applicable to any absorption values. You can easily check this by the limit of strong absorption:
when $\alpha = \infty$ and $L(\alpha) = 0$, therefore $1-\omega = T_{out}(n)$ or $\omega = 1-T_{out}(n) = R_{out}(n)$, which means that all the light that goes into the particles is absorbed. This limit is not true for A2. For comparison you'd better take eq. 25 for small absorption instead of general eq. 23: $1-w = n^2 \alpha\mu_1$ (in your notation) and easily find the B-factor $B = n^2 = 1.68$ at 1.3 um for ice. The deviations of B from this value demonstrate the difference between the models used by Libois et al. and the model of the random mixture.

**Reply:** We agree that the limiting case of $\alpha$ and to $\infty$ is not consistent in both expressions. However in practice we compare both expressions only in the limit of small $\alpha$, for which both are supposed to be valid. This issue was brought up also by the second referee under point 2 and is further discussed there.

**Changes to the manuscript:** We clarified the underlying assumptions in the appendix and added necessary details to the discussion of the Figure in the discussion section.

References by the referee:

P. Switzer, "A random set process in the plane with a Markovian property," Ann. Math. Statist 36, 1859-1863 (1965).

References by the authors:

Calonne, N., Flin, F., Lesaffre, B., Dufour, A., Roulle, J., Puglièse, P., Philip, A., Lahoucine, F., Geindreau, C., Panel, J.-M., Rolland du Roscoat, S., and Charrier, P.: CellDyM: *A room temperature operating cryogenic cell for the dynamic monitoring of snow metamorphism by time-lapse X-ray microtomography*, Geophys. Res. Lett., 42, 3911–3918, doi:10.1002/2015GL063541, 2015.

Löwe, H. and Picard, G.: *Microwave scattering coefficient of snow in MEMLS and DMRT-ML revisited: the relevance of sticky hard spheres and tomography-based estimates of stickiness*, Cryosphere, 9, 2101–2117, doi:10.5194/tc-9-2101-2015, 2015.

Löwe, H., Spiegel, J. K., and Schneebeli, M.: *Interfacial and structural relaxations of snow under isothermal conditions,* J. Glaciol., 57,499–510, doi:10.3189/002214311796905569, 2011.

Löwe, H., Riche, F., and Schneebeli, M.: *A general treatment of snow microstructure exemplified by an improved relation for thermal conductivity*, The Cryosphere, 7, 1473–1480, doi:10.5194/tc-7-1473-2013, 2013.

Mätzler, C.: *Relation between grain-size and correlation length of snow*, J. Glac., 48, 461–466, doi:10.3189/172756502781831287, 2002.

Proksch, M., Mätzler, C., Wiesmann, A., Lemmetyinen, J., Schwank, M., Löwe, H., and Schneebeli, M.: MEMLS3&a: *Microwave emission model of layered snowpacks adapted to include backscattering*, Geosci. Mod. Dev., 8, 2611–2626, doi:10.5194/gmd-8-2611-2015, 2015b.

Vallese, F. and Kong, J.: *Correlation-function studies for snow and ice*, J. Appl. Phys., 52, 4921–4925, doi:10.1063/1.329453, 1981.

---

## Author Comment (AC2) · 3 Sep 2016

**General comments:**

This paper addresses the relation between the grain metrics commonly used to model snow optical and microwave properties. At first order, snow microwave properties are governed by the exponential correlation length $\xi$ while snow optical properties firstly depend on snow specific surface area (SSA). However, at second order snow grain shape also affects snow radiative properties. From this statement, statistical relations are derived that make the link between snow microstructure characteristics (curvatures) and snow physical properties. The relation between $\xi$ and SSA is thus improved compared to previous empirical relations, by adding a contribution of snow grain shape. The general theoretical framework of Malinka (2014) is then used to show that snow optical properties depend on the moments of the chord length distribution. Based on this framework, another statistical relation is derived to express the second moment of the chord length distribution in terms of microstructure length scales. From this, a statistical relation between $\xi$ and the first two moments of the chord distribution is derived. This suggests that shape parameters derived from optical measurements could be used as inputs for snow microwave modeling. This point is supported by comparing the values of the optical shape parameter B deduced from Malinka (2014) theory to values determined experimentally.

The paper is overall well written and pleasant to read, the objectives are well defined at the end of the introduction. The theoretical background is nicely presented and clearly outlines the problem. The approach is original and takes advantage of recent works in snow optics. It also applies the statistical properties of general random heterogeneous materials to the case of snow, thus linking rather theoretical studies and practical cases as illustrated by the use of μCT images of snow samples. The authors stress the need for a unified definition of grain shape and propose mean curvatures as such definition. They show that both microwave and optical properties can be expressed in terms of SSA and mean curvatures. Their approach is supported by the analysis of a large set of μCT images. They also provide valuable physical insight on the representation of snow microstructure as a particulate or heterogeneous medium. For these reasons, I recommend this paper be published in The Cryosphere. However, a number of critical points should be addressed before publication, related in particular to the fundamental assumptions underlying the presented theoretical framework.

**Specific comments:**

The theoretical framework presented in this paper strongly relies on critical assumptions that are not sufficiently discussed, although several important results largely depend on them.

1) Throughout the text, snow is considered isotropic and the derivations significantly rely on this critical assumption. Although this assumption is clearly stated, several details are lacking to convince the reader that the results remain reliable. First, more details on the investigated snow samples should be provided. So far only 3 lines (section 3.1) present these critical

elements of the study, which is not enough. Do these samples consist of sifted snow, natural snow samples taken in the field without perturbing the microstructure, snow samples resulting from metamorphism experiences in the laboratory...? It is clear that depending on the origin of the samples, the isotropic hypothesis is more or less acceptable. For instance depth hoar is known to be highly anisotropic and can hardly be investigated under this hypothesis. The authors should consider removing highly anisotropic snow samples if they do not fit in the theoretical background.

At the same time, the authors do have the necessary material to further discuss the isotropic hypothesis because the parameters are obtained from averages over the 3 directions x, y and z. Giving a hint of the actual anisotropy from the analysis of these 1-D parameters might help the interpretation of the data and estimate the associated uncertainties.

**Reply:**
We probably did not sufficiently elaborate on that point. To begin with, it is important to note that, strictly speaking, our analysis does not *assume* isotropy. We rather employ (wherever necessary) orientational averaging to reduce the information that is eventually used for the analysis. The geometrical interpretation of the involved quantities does not rely on isotropy. As an example, the relation between the slope of the correlation function via $\lambda_1$ and the surface area hold also (rigorously) for arbitrary, anisotropic systems, *after* orientational averaging (Berryman.1987). The same is likely true also for $\lambda_2$ , namely that the orientational average of the third derivative of the correlation function of an anisotropic system is related to interfacial curvatures in the suggested way. We did not find a mathematical proof of the latter statement in literature, but our comparison of $\lambda_2$ (obtained from the correlation function, orientationally averaged) with $\lambda_2$ (obtained from direct computation of the interfacial curvatures) strongly suggests its validity. As an additional confirmation, we checked (plot below) that the remaining scatter is not caused by anisotropy, by plotting the residuals between the estimate $\lambda_2^{vtk}$ (where anisotropy does not play a role) and $\lambda_2^{cf}$, which is not correlated with the anisotropy ($R^2$=.026). Accordingly, we also use the other length scales in the meaning of orientational averages, of arbitrary anisotropic systems. For the exponential correlation length this has been done similarly before. That said, none of the samples must be discarded.

Criticality of this procedure (not assumption) can only be revealed by measurements that will decide about the relevance of these orientationally averaged length scales for a measurement of anisotropic nature.

**Changes to the manuscript:** We add a part to the Discussion that discusses the anisotropy and retrieval of the parameters, however without showing this plot.

[Figure]

2) In this study, the successive chords in snow are assumed independent, which is a strong assumption not really defended by the authors. This same assumption was used by Malinka (2014) who considered a random medium, whose optical properties where then derived.

However, this author clearly states in his conclusion that: "The requirement of stochasticity is mandatory: the facets orientation and the ray path length inside solid or voids must be independent variables. [...] The question of applicability of the model to any particular medium should be considered separately based on compliance with the experimental data." Practically, one might expect light rays to be trapped in snow grains or selectively focused in preferential location, which would result in different chords having different realization probabilities.

A critical consequence of the random distribution hypothesis is that at low ice absorption, the optical properties of this random medium do not depend on the shape parameter (see e.g. eq. (25) of Malinka (2014) from which B can easily be derived). This is somehow contradictory with the definition of grain shape, which is expected to impact snow optical properties in the standard particular representation.

An alternative approach could be to validate this random medium assumption by comparing the values of B retrieved from Malinka (2014) to those determined by Libois et al. (2014), which are very similar. Once the random medium hypothesis is somehow validated, then the shape parameter only impact optical properties at more absorbing wavelengths. An important corollary of this would be that only optical measurements at relatively absorbing wavelengths would contain information about snow grain shape.

**Reply:**

We agree that for this step of deriving the shape parameter B from Libois 2013 by using Malinka 2014 involves a particular assumption about the independence of chords and adjacent surface normal orientations (note that our analysis of the statistical links between the length scales is however not affected by this) This issue was also brought up by the other reviewer, however rather pointing out the assumption of low absorption underlying Libois 2013 (which in contrast does not affect Malinka 2014). We are thus faced with the situation of linking two models/expressions that are based on two different, disjunct assumptions. That said, it is not entirely correct of using the closeness of the values found here to the values from Libois 2014 to confirm that the assumption of independent chords is not very restrictive. This aspect is now explained in more detail when deriving and discussing this connection. In the end (to produce Fig8) we evaluate B in the limit of low absorption (to cope with Libois). It is important to note that the assumption of independent chords (used in (Roberts and Torquato) already mentioned now in the paper) is slightly different from the assumption used by Malinka 2014. This will be also made clear.

**Changes to the manuscript:** The derivation of B is extended $g^G$, and the assumptions are discussed.

3) When it comes to the analysis of μCT images, the question of voxel size (ie resolution) is not enough discussed. In fact, the resolution varies from a set of measurements to another and is generally not that small compared to snow size metrics. This probably has an effect on the derived results and might explain why different subsets of points appear on several Figures (e.g. 2 bottom left and 4a). The smoothing parameter is discussed in sufficient details but resolution is probably an issue as critical.

**Reply:** We agree that the possible impact of the resolution could influence the results if the obtained quantities of interest are within a similar range. In general, the choice of resolution for CT images is made in accordance with the structure, such that the sample/resolution is statistically representative for the main quantities of interest (density and specific surface area). To assess the reliability of the obtained results we have compared them to the alternative VTK based method, for which we find very similar results. If the values for s are compared to the values that are obtained by the vendor software, we also see a good agreement with the VTK based method. To further confirm that the main quantity $\lambda_2$ is not systematically affected by image resolution we have plotted below the ratios of $\lambda_2$/voxelsize as a function of voxelsize, which are on average 9.8 with a standard deviation of 2.6. Only two

samples have ratios 4.5 and the rest is 6.0 and higher. The correlation with the voxel size is $R^2$=-.20, but overall there is no systematic trend in $\lambda_2$/voxelsize for lower resolution (which would indicate a worse representation of the characteristic scales).

[Figure]

**Changes to the manuscript:**
We added a sentence on the spatial resolution of the data sets, its general importance and added the values for the characteristic ratios $\lambda_2$/voxelsize (the plot is however not included)

4) Although snow optical properties equally depend on the parameters B and g, the paper is mostly focused on B. The analysis presented for B can very easily be extended to g. This would be more exhaustive because all parameters relevant to snow optics would be tackled, as all parameters (actually only $\xi$) needed for snow microwave modelling are.

**Reply:** We agree that this extension to g (or $g^G$) is worthwhile for a comparison to Libois.2013. We replaced figure 8 by a plot of 1-$g^G$ versus B (similar to libois.2013).

**Changes to the manuscript**: Table 1 is extended and Fig.8 is replaced by the a plot of B versus 1-$g^G$

5) The manuscript would benefit from a slight reorganisation of some parts because redundancy is found at several points and excessive details sometimes pollute the paper. Some elements are given too early (e.g. details about the Euler characteristic that should probably not be mentioned before the discussion section), some others should be provided in a different order (more details are provided along the technical comments). Also sections 3.3 and 3.4 could probably be merged.

**Reply:** We agree, this is also in accordance with a suggestion of the editor. As suggested, the definition of the Euler characteristic in the theory section is left out left out, since it is not explicitly required. The Discussion section is restructured. It discusses first the methodology, including resolution, anisotropy, and the geometrical interpretation of $\lambda_1$ and $\lambda_2$. Afterwards, that the statistical models are discussed. We finalize it by discussion grain shape, including the connection to (Libois.2013,2014) and (Malinka.2014).

**Changes to the manuscript:** As indicated above.

6) The authors make their best to infer the shape of the statistical relations from theoretical backgrounds. However, this often adds noise to the paper because 1) the underlying assumptions are often very restrictive and not applicable to snow (dilute medium, random medium, use of Taylor expansion at 0 for estimating functions at infinity...) and 2) these statistical relations are eventually revisited by adding terms. I think there is no problem assuming a certain type of relation, and then testing it with the available data. For sure, the type of relation can be suggested by a rapid analysis of existing formulae, but there is no need trying to justify it too much. In this context, I would suggest to remove the unnecessary calculations and reformulate the section around Eqs. (14) and (15). For instance the authors could say that they show the validity of Eq. (14) from images, even though initially this relation is only valid to restricted cases. All the attempts to justify this equation are unnecessary

**Reply:** We agree. This is also in accordance with the other reviewer. These points are left for the discussion.

**Changes to the manuscript:** The motivation for eq.15 is removed and this section is reformulated.

7) The authors should give a consistent name to all important quantities $\xi$, $\lambda_1$, $\lambda_2$, $\mu_1$, $\mu_2$ and keep those names all along the manuscript. For instance, exponential correlation length and correlation length are sometimes used alternatively without a clear distinction. Porod length, optical diameter and curvature length are used sporadically as well.

**Reply:** This was basically an attempt to stick to the names previously used in literature. But we agree, naming is now consistent and less ambiguous: $\lambda_1$ is named the Porod length, $\lambda_2$ is named the curvature length, the name for $\xi$, the exponential correlation length, remains. For $\mu_1$ and $\mu_2$ we stay with the first and second moment of the chord length distribution.

**Changes to the manuscript:** The naming is made consistent throughout the manuscript.

8) At the light of the comments above, it will probably be necessary to rewrite the last section of the discussion (5.4).

**Reply:** We agree, see comment 5).

**Changes to the manuscript:** The discussion rewritten, taking all comments from both referees into account

**Technical comments:**

Could "snow grain size" be used instead of "grain metrics of snow"? Alternative suggestions (these are only suggestions):
- "Relating optical and microwave snow grain size: The importance/relevance of using/considering grain shape"
- "Accounting for snow grain shape to improve the relation between optical and microwave snow grain size"
We agree (maybe) to be discussed.

Abstract:

p.1 l.1: rephrase to better compare the roles SSA and exponential correlation length play in determining snow optical and microwave properties. Either from the physical point of view: "microwave emissivity/properties mostly depend(s) on the exponential correlation length". Or from the modelling point of view : "the exponential correlation length is the relevant quantity in most snow microwave models" or "the exponential correlation length is used to simulate snow microwave properties"
**Reply:** We agree.
**Changes:** The sentence is changed to "the exponential correlation length is the relevant quantity in most snow microwave models".

p.1 l.3: a microwave model is not "forced" by optical measurements, it uses quantities derived from optical measurements (e.g. SSA) as inputs. Forcing more generally refers to something external to the system (e.g. boundary conditions). This is correctly said p.2 l.9.
**Reply:** We agree.
**Changes**: "To facilitate forcing of microwave models by optical measurements" is replaced by "to derive input quantities of microwave models from optical measurements".

p.1. l.3: "the understanding of $\xi$" is vague. Simply say "To refine this relation between...]"
**Reply:** We agree.
**Changes:** the sentence is adjusted to "To refine the relation between…"

p.1 l.5: it is a statistical relation more than a prediction
**Reply:** We agree.
**Changes:** "Prediction" replaced by "relation".

p.1 l.8-9 : maybe remove this sentence because it does not provide additional information about the results. Also, it is somehow questionable in terms of applicability within the present theoretical framework. Keep it for the body of the manuscript.
**Reply:** We agree.
**Changes:** Deleted.

p.1 l.10 : B is called the absorption enhancement parameter. Consider doing the same calculations with g.
**Reply:** We agree.
**Changes:** The parameter g, and therefore $g^G$, can be directly inferred from (Malinka.2014). This is added to the analysis and abstract.

p1. l.10 : the last sentence of the abstract is not clear. Maybe say "Our results suggest that optically derived shape parameters can be used to refine the estimation of $\xi$".
**Reply:** We agree.
**Changes:** Last sentence changed to say "Our results suggest that optically derived shape parameters can be used to refine the estimation of $\xi$".

Introduction

p.1 l.16-19 : maybe invert the order of the two sentences to keep chronological order
**Reply:** We agree.
**Changes:** The sentences are inverted.

p.2 l.4 : "with the MEMLS model" instead of "is used"
**Reply:** We agree.
**Changes:** Adjusted accordingly.

p.2 l.14 : "though less significant..." is risky because the impact can actually be significant (errors up to 50%) for BRDF or light penetration simulations for instance.
**Reply:** We agree.
**Changes:** Changed.

p.2 l.16 : reference to Picard et al. (2009) might be relevant
**Reply:** We agree.
**Changes:** Reference is included.

p.2 l.17 : in this study the absorption enhancement parameter B and asymmetry factor g (name these factors) are equally important, except that only B can be estimated from optical measurements. Note that Libois et al. (2014) experimentally determined the parameter B for a variety of natural snow samples.
**Reply:** Thanks for pointing this out; we have not been aware of the paper.
**Changes:** Sentence on the measurement of B is added, including the citation. The discussion of B comes back to this point.

p.3 l.1 why "systematically?"
**Reply:** No specific reason.
**Changes:** Systematically is deleted.

p.3 l.12 : not clear what "images" you're talking about
**Reply:** We agree.
**Changes:** "images" is replaced by "μCT images".

p.3 l.15-17 : maybe keep those last 2 sentences for the discussion and mention it more shortly at this stage because this is hard to understand without the whole paper in mind.
**Reply:** We agree.
**Changes:** Rephrased

Theoretical Background

p.3 l.21-22 : very redundant with p.1 l. 20-21.
**Reply:** We agree.
**Changes:** The sentence has been reformulated.

p.4 l.5 : why "in contrast"? Is the exponential approximation only valid for large r values?
**Reply:** We agree. The exponential approximation is of course based on a fit for *all* r.
**Changes:** We removed "In contrast" from the sentence.

p.4 l.14: use m2 kg-1 instead
**Reply:** We agree.
**Changes:** Adjusted accordingly.

p.4 l.24-28 : consider mentioning the topological dimension of the mean Gaussian curvature only in the discussion, because at this stage the reader does not understand the point.
**Reply:** We agree.
**Changes:** Removed and included in the discussion.

p.4 l.26: the mathematical notation is not clear. Maybe use dS or dA to explicitly state that this is an average on the surfaces? This integration element could also be moved after the integrand.
**Reply:** We agree. The reference to the Euler characteristic is however moved to the discussion.
**Changes:** Adjusted accordingly.

p.4 l.27: that the local. Why is local in parenthesis?
**Reply:** Local refers to the fact that the determination of this part of the correlation function is an average over nearest (or next nearest) neighbours (in the voxel images) which is commonly referred to as "local". This is contrasted non-local (i.e. long range) effect.
**Changes:** The sentence is moved to the discussion, and local is removed to avoid confusion.

p.6 l.10: detail why z is actually small and mention in which conditions this theoretical framework is valid. This in in fact detailed below, but inverting the order might be helpful.
**Reply:** The text could indeed improve from reordering these sentences.
**Changes:** Reordered accordingly.

p.6 l.13 : to the theory of
**Reply:** We agree.
**Changes:** 'the' is inserted.

p.6 l.14 : it's $4\pi$ rather than $2\pi$.
**Reply:** We agree.
**Changes:** Changed accordingly.

p.6 l.20: state here that the following sections investigate this issue and try to find a geometrical meaning of this second moment.
**Reply:** We agree.
**Changes:** A sentence is inserted.

p.7 l.2 : would it be useful to briefly define the surface-void correlation function? Otherwise
**Reply:** We won't go into the precise definition of the surface-void correlation function since it does not affect the understanding of the method. It seems however justified to mention it here since this part indicates the required effort to improve the relation between the two point correlation function and the chord length distribution to be valid not only for dilute systems (comment from the other reviewer).
**Changes:** No

p.7 l.4 : please clarify the meaning of "this is not a practical limitation"
**Reply:** This question is related to the more fundamental question about the validity of independent chords from point 2.
**Changes:** see point 2.

p.7 l.1-7: since eventually the relation of Roberts and Torquato (1999) is not used, this part adds noise to the paper. Consider removing it (or mention it more concisely) if indeed it is not used.
**Reply:** We agree that we did not exploit this reference extensively. It is however crucial to comment on the assumption of the independence of consecutive chords.
**Changes:** The sentence is reformulated and used for a slightly different purpose (addressing point 2).

p.7 l.12: not clear why you keep going while snow is clearly not a dilute medium. If the relation actually holds for snow (which seems to be the case as you show its consistency), state there that you demonstrate its validity for snow.
**Reply:** We agree. (see also point 6). This point has been left out here since we come back to it in the discussion.
**Changes:** The section is cleaned up accordingly.

p.7 l.15: it seems that integrating by parts result in a factor [dA(l)/dl ]. Why is it equal to 0? True for the exponential case. Idem for p.7 l.18
**Reply:** The two-point correlation (and thus its derivative) must go to zero for random systems for large r. Only in the presence of long range order (e.g. objects placed on a regular lattice) correlations persist to infinity (periodicity)
**Changes:** None.

p.7 l.20 : the expansion is only valid for small r values, while here the integration goes much beyond.
**Reply:** This equation is removed in the new manuscript, and therefore not discussed here anymore.
**Changes:** Revision of page 7.

p.7 l.20-24 : This paragraph somehow adds noise to the flow of the paper. Would it be problematic to make it shorter and simply state that in Eq. (15) the integral is a function of $\lambda 1$ and $\lambda 2$ and must be of "length" dimension? I think this would not change the use of this equation later on (section 4.4). This approach would also allow the use of a constant term in the fit of Eq. (21) without further justification.
**Reply:** We agree. Thank you for this suggestion, which serves as the basis for the new formulation.
**Changes:** Revision of page 7.

Methods
p.8 l.4 : More details about the preparation of the samples should be provided, and the isotropy of the prepared samples should be discussed. If for instance some samples obviously do not follow the

isotropy requirement (e.g. depth hoar) they should be removed from the analysis.
**Reply:** We agree that we could include more information on the samples that are used. Next to that, the isotropy (or rather absence of it) is mentioned. In the discussion session this is treated more extensively.
**Changes:** More information on the samples is given, and isotropy is shortly discussed.

p. 8 l.10 : the point regarding voxel size is very critical because the length scales are similar to voxel size, implying potential impact of voxelisation on the results. Can images at 18 and 50 μm be compared? See specific comment 3.
**Reply:** See answer to Comment 3.
**Changes:** We have discussed the effect of resolution in the methods and we come back to that in the discussion in more detail.

p.8 l.11 : before averaging, an evaluation of the anisotropy (or isotropy) should be given, because the whole theoretical framework is based on the isotropic hypothesis.
**Reply:** see answer on point 2
**Changes:** see comment 2.

p.8 l.15 : Figure 1b does not really illustrate the exponential regression
**Reply:** In fact the formula that is used to create this figure is an exponential function. The illustration is a graph representing the involved parameters.
**Changes:** Figure is adapted with an illustration of the retrieval of $\lambda_1$ and $\xi$.

p.8 l.23 : the meaning of "in view of shape" is not clear.
**Reply:** We agree.
**Changes:** This sentence is changed to: "To confirm the geometrical interpretation of $\lambda_1^{cf}$ and $\lambda_2^{cf}$ we use an alternative and independent method to estimate these parameters by measuring the surface area and the local interface curvatures with a VTK-based image analysis. In short… "

p.8 l.23-25 : state more clearly that the section aims at validating the Eqs (6) and (8) by computing the interfacial area and interfacial curvatures.
**Reply:** We agree.
**Changes:** see previous changes.

p.8 l.30 : could this smoothing parameter be slightly more detailed, because it seems critical in the following section. What's the typical range, what values were used in the past? For what kind of applications?
**Reply:** We agree. The smoothing parameter is a value for the number of times the Laplacian smoothing operation is applied. The smoothing has been discussed in (Krol.2016) and we adopted the same value for S here.
**Changes:** A short description of the filter is added.

p.9 l.4 : for S = 200, the interfacial area is larger, but the points seem also more spread, which is not discussed.
**Reply:** This is true. This is likely due to the fact that smoothing is filtering out small perturbations in the surface, reducing the area and increasing the values for $\lambda_1$. To which extend this happens is sample dependent, which causes the estimate for $\lambda_1$ to show a higher variance.
**Changes**: A sentence is added to clarify this.

p.9 l.6-11 : what is the objective of this section? Does it serve the paper? Should it be used to support the isotropic hypothesis?
**Reply:** We partly agree. We removed the figure but we kept this small paragraph to elaborate more on the surface representation and smoothing. The factor of 3/2 has been the origin of quite some confusion in the past, and we would like to take the opportunity to mention and hopefully clarify this point.
**Changes:** Figure is removed.

p.9 l. 16 : one should be with superscript "cf"
**Reply:** We agree.
**Changes:** adjusted.

Figure 2 (bottom left) : there seems to be 2 sets of points, one consisting of RG. Could this observation help interpreting the limitation of S = 50?

**Reply:** In fact there are as many 'groups' of data as there are time-series present, which naturally show a pseudo- continuous deviation from the curvature estimates. The deviations from the 1:1 line are caused by the overestimation of the curvatures by the remaining steps in the triangulation from the underlying voxel-based data, and is thus anti-correlated with the size of the structures and correlated with voxel size. In the end we chose a smoothing parameter that is, on average, acceptable for all involved samples.

**Changes:** A sentence is added to the discussion to clarify this apparent grouping of samples.

Figure 4a : there seems to be 2 sets of points. Do they correspond to similar subsets of µCT images? The same 2 sets are observed in Fig. 6a

Figures 4b and c : DH is clearly an outsider here. Is it relevant to keep it in this study?

**Reply:** As explained above, these two sets of points are correlated since they are part of a time series. We will emphasize this when the samples are introduced in section 3.1. The depth hoar samples that show a higher deviation in Fig 4b and 4c, do not have particularly higher anisotropy values than the other depth hoar samples that do not have high residuals.

**Changes:** The data is introduced in more detail as well as the fact that some of them are part of a time-series. The anisotropy is discussed in more detail in the reformulated discussion.

Results

p. 11 l.11 : one extra "and"
**Reply:** We agree.
**Changes:** 'The' is deleted.

p. 11 l.11 : is it consistent to have a R2
 less (0.731<0.733) for the regression with an additional
parameter?
**Reply:** Yes it is, since fitting eq.(18) includes two extra parameters which, if done correctly, should be accounted for in an adjusted correlation coefficient. Since $a_0$ is negligible to the fit this does not show in $R^2$ but it is however penalized in the reduced correlation coefficient.
**Changes:** we included "adjusted correlation coefficient".

p.13 l.1 : the name of λ1 should be consistent between titles of sections 4.1 and 4.2. In section 4.1, optical diameter is not mentioned except in the title.
**Reply:** We agree.
**Changes:** The subtitle is changed from "Relating exponential correlation length to optical diameter" to "Relating exponential correlation length to the Porod length".

p.13. l.7 : I don't really understand this justification and don't think this is necessary. I would proceed the other way round instead. The figure 4b could be discussed at the end of section 4.1 with the aim of understanding the remaining residuals. This would naturally lead to the regression Eq. (19).
**Reply:** We agree.
**Changes:** The order is reversed.

p.13 l.13 and 14: Eq. (14) instead of (16)
**Reply:** We agree.
**Changes:** Adjusted.

p.14 l.3 : Eq. (15) in stead of Eq. (14)
**Reply:** We agree.
**Changes:** Reference adjusted.

p.14 l.17 : here you try "heuristically" a regression, which is fine. This somehow contrasts with the previous regressions that were based on the derivation of equations. This could also be motivated by the form of Eq. (13) that includes the porosity factor. I think there is no problem assuming a relation,

and then testing its validity with measurements. This is sometimes easier to understand than long inexact derivations.

**Reply:** We agree. In this section we changed the motivation for the statistical models involved.

**Changes:** We reformulated this sentence to "To motivate a statistical model we start from eq.(15) and test different expressions for $f(\varphi,\lambda_1,\lambda_2)$. Since f has dimension length a natural first candidate would be.. ". l.8: The sentence "Although not predicted from Eq.20…" is removed.

p.14 l.12 : it is awkward to read that the benefit is small but to see the new regression, though. I would put it more positively: "The correlation coefficient (R2=0.295) is small but including λ2 in the analysis further improves the fit".

**Reply:** We agree.

**Changes:** adjusted

p.14 l.24-25 : this is sometimes disturbing to read "correlation length" at some point and "exponential correlation length" later on. Please remain consistent throughout the manuscript, with each quantity ($\xi$, $\lambda_1$, $\lambda_2$) having its dedicated and constant name. Consider using "exponential" for the first part of the sentence, and "correlation length scales or Porod length and curvature lengths (for instance)" for the second part, to make the link with Eqs. (19) and (23) more obvious.

**Reply:** See comment 7. To avoid confusion between the exponential correlation length and correlation length we stick to the term Porod length for $\lambda_1$.

**Changes:** Naming changed.

Figure 6 : remove "see". λ1 is not the optical diameter.

**Reply:** We agree.

**Changes:** Removed "optical diameter", changed to Porod length.

Discussion

p.16 l.2 : in complement to this discussion, this might be worth giving the sensitivity of Eq. (16) to the smoothing parameter, and possibly to the voxel size as well, if this makes sense.

**Reply:** The smoothing parameter only influences the VTK-based parameters. The voxel size has an impact on the estimates of $\lambda_1$, $\lambda_2$, $\mu_1$ and $\mu_2$ and will be discussed in more detail.

**Changes:** Voxel size is detailed in the discussion.

p.17 l.5 : remind what grain size is because a1 is the coefficient for λ1 (which is optical diameter or grain size?)

**Reply:** We agree.

**Changes:** We adapted this sentence to "As a first step we have analysed the statistical relation between exponential correlation length and the Porod length. The latter is referred to as simply "grain size" or correlation length in Mätzler( 2002)".

p.17 l.6 : again depth hoar could be removed from the analysis if it does not satisfy the conditions of the theoretical framework.

**Reply:** As discussed under point 1. Accordingly, we will argue in favour of keeping these samples.

**Changes:** None.

p.17 l.7 : this is not clear what is also shown by those data. That the coefficient is larger for depth hoar?

**Reply:** The results from Mätzler( 2002) also distinguish depth hoar $\xi=.8\lambda_1$ and other snow types $\xi=.6\lambda_1$.

**Changes:** The sentence is changed to "Mätzler's model predicts a1 = 0.75, which is an average of $a_1=.8$ for depth hoar and $a_1=.6$ for other snow types. Comparing this to our result, $a_1=.79$, this is consistent since we have many depth hoar samples in the data set, which indicates an even larger influence of snow type or grain shape."

p.17 l.21 : Eq. (7) instead of Eq. (1)

**Reply:** We agree.

**Changes:** adapted

p.17 l.32 : there were attempts

**Reply:** We agree.
**Changes:** adapted

p.18 l.5 : why is "independent" in italic. Idem for p.18 l.15 "if"
**Reply:** We agree that it is not necessary to stress the words 'independent' and 'if'.
**Changes:** Adapted.

p.18 l.5 : where does this K/3 come from? It is K/24 in Eq. (8)
**Reply:** Yes, K/3 must be compared to $H^2$.
**Changes:** We adapted Eq(8) to $1/8( H^2-K/3)$ to make this obvious.

p.18 l.12 : this point is interesting, but puzzling as well. Indeed, from an optical point of view, a polydispersion of spheres will have the same "shape" parameters as a monodispersion in the geometrical optics approximation (and for low ice absorption), because B and g primarily depend and the shape, not on the size. Hence polydispersion would affect curvatures, but not grain shape as defined from an optical point of view. Said differently, a polydispersion of spheres will have optical properties similar to a monodispersion with same SSA, but different microwave properties.
**Reply:** We agree.
**Changes:** None.

p.18 l.32 : for such a system?
**Reply:** Yes.
**Changes:** Changed.

p.19 l.10 : wavelengths (in a single word?)
**Reply:** We agree.
**Changes:** wave lengths -> wavelengths.

p.19 l.12 : the mentioned paper rather suggests that g for spheres is larger than g for snow, and that B for spheres is smaller than B for snow.
**Reply:** We agree. This is also consistent with the values we calculated for g and B shown now in Fig.7
**Changes:** Adapted.

p.19 l.12 : the superscript G for the g refers to "geometrical", that does not account for the diffraction contribution to scattering. This does not change the sentence but should remain consistent throughout the paper.
**Reply:** We will use consistently $g^G$ and B.
**Changes:** notation adapted.

p.19 l.12 : it depends on shape rather than includes it
**Reply:** We agree.
**Changes:** include->depends

p.19 l.16 : it's $4\pi$ rather than $2\pi$. By the way this quantity was already defined p.6. Then check the values for the following text and those shown in Table 1.
**Reply:** Checked
**Changes:** None.

Table 1:

Fraction of second to first rather than first to second order. Precise that mean and standard deviation are among all samples. Write 170 rather than $1.7 \times 10^2$.
The values suggest no influence of shape at 0.9μm, which is consistent with the remark p.18 l.12.

Note that eq. (5) of Malinka (2014) shows that at weakly absorbing wavelengths, B only depends on the real part of the refractive index.

This latter point should be further discussed to explore the validity of the random medium assumption used by Malinka (2014). In fact, this framework suggests that as long as the structure is random, shape

has no impact on optical properties. This is contradictory to the fact that in the particulate representation of snow, different grain shapes result in different optical properties, even at low ice absorptions

**Reply:**
We adapted the notation and description in Table 1.

We agree that Malinka involves a particular assumption on the independence of chords and adjacent surface normal orientations. This apparently leads to $B=n^2$ in the limit of very small alpha. This is now explicitly shown in the appendix. We also calculated there the next order correction in alpha that shows a slight dependence of B on shape if the latter would be defined only via moments of the chord length distribution. Accordingly, for visible wavelengths and corresponding alpha, no shape dependence of B would be predicted from A4, which is indeed not what is observed in nature. Thus it might be the case that, by using this independence assumption, some influence of shape on B is lost, in particular for for very low alpha (visible).

**Changes:** We included these points in the Discussion.

p.20 l.6 : the authors decide to emphasize the parameter B, but in fact eq. (60) of malinka (2014) can also be used to express g in terms of λ1 and λ2. This should be done to complete the analysis.
**Reply:** We agree. The analysis is extended to g
**Changes:** New figure with a plot of g versus B.

p.20 l.7 : why is the parameter B shown in terms of this ratio? Is there supposed to be a visual correlation in Fig. 8? Why is the regression with respect to this particular ratio?
**Reply:** This was done because eq.A4 is a function of p(α), and the ratio determines the relative importance of first and second order terms. However this figure is now replaced by a plot of g versus B. But the ratio can be also used as a simple proxy to assess the deviation of the snow chord length distribution from an exponential one (see comment 6 in the other review) Values are therefore given in the text.
**Changes:** Figure changed.

p.20 l.9 : Libois et al. (2014) experimentally determined the parameter B for a large set of snow samples and suggest B equals $1.6 \pm 0.2$. This comparison completes that with Libois et al. (2013). Note again that the range obtained in Fig. 8 results from the impact of shape at 1.3μm. This range can hardly be compared to that obtained by Libois et al.(2013,2014) obtained at visible wavelengths. The absolute values can on the contrary be compared.
**Reply:** We will emphasize the difference in the wavelength and discuss that in the weakly absorbing limit B is only depending on the real part of the refractive index. We will also point out that the apparent increased variation of B observed for visual wavelengths, may be due to the shadowing effect/density as discussed in (Libois et al (2014).
**Changes:**

p.20 l.9-12 : these sentences are not clear, and reference to Haussener et al. (2012) is very fuzzy, in particular the "remaining discrepancies".
**Reply:** we agree.
**Changes:** reference removed

p.20 l.15 : involved
**Reply:** We agree.
**Changes:** adapted

p.20 l.20 : this is the very critical assumption that should be further discussed
**Reply:** We agree. This assumption is indeed critical, but rather difficult to investigate. As explained above, we can only discuss this in reference to (Roberts and Torquato) who established an improved relation between the chord length distributions and the correlation functions. Their improved relation is still based on the assumption of independent chords. They tested this for level-cut Gaussian random fields, where successive chords are not independent from a rigorous perspective. The results however agree reasonably well, which is at least an indicator that this assumption is not so critical for this

aspect. As mentioned before, this independence assumption is however still slightly different from the independence assumption used by Malinka 2014.
**Changes:** This point is emphasized in the discussion which has been restructured.

p.21 l.l.1-16 : This part shows is partly redundant with previous parts of the text. This could be shortened.
**Reply:** This part of the text is replaced and rewritten to avoid redundancy.
**Changes:** Discussion is restructured and rewritten.

p.21 l.11 : why is this work mentioned here and not before? Could this help to establish the semiheuristical relations displayed all along the manuscript?
**Reply:** This relation is introduced in the discussion since it only explains that the slope in the origin of the chord length distribution is related to $\lambda_2$. While this shows yet another connection between chord lengths and the curvature lengths, worth mentioning, we were not able to put this on more general grounds which could be exploited earlier.
**Changes:** None.

p.21 l.12-14 : Why is the variance of the chord length distribution mentioned here for the first time?
**Reply:** Because it emerges only here in this argument to connects mu2 to lambda_12
**Changes:** In the reformulated discussion the variance of the chord length distribution is left out.

p.21. l.19 : remove parenthesis in reference
**Reply:** We agree.
**Changes:** removed

Conclusions

p.21 l.29 : extra "we"
**Reply:** Yes.
**Changes:** Changed.

p.21 l.29 : consider adding ($\lambda2$) after size metric
**Reply:** We agree.
**Changes:** added

p.22 l.9 : the meaning of "when compared to" is not clear
**Reply:** we agree.
**Changes:** rephrased.

p.22 l.9 : Maybe say : "The consistency between B values derived from the chord length distribution and those determined from optical measurements suggests such an approach is indeed possible".
**Reply:** We agree,
**Changes:** Changed accordingly.

Appendix

p.22 l.28 : no parentheses for the references
**Reply:** We agree.
**Changes:** parentheses removed

p.23 l.8 : by the Swiss...
**Reply:** we agree.
**Changes:** The typo is removed.

References by the referee:

Haussener, S., Gergely, M., Schneebeli, M., & Steinfeld, A. (2012). Determination of the macroscopic optical properties of snow based on exact morphology and direct pore- level heat transfer modeling.

Journal of Geophysical Research: Earth Surface, 117(F3).

Libois, Q., Picard, G., Dumont, M., Arnaud, L., Sergent, C., Pougatch, E., ... & Vial, D. (2014). Experimental determination of the absorption enhancement parameter of snow. Journal of Glaciology, 60(222), 714-724.

Malinka, A. V. (2014). Light scattering in porous materials: Geometrical optics and stereological approach. Journal of Quantitative Spectroscopy and Radiative Transfer, 141, 14-23.

Picard, G., Arnaud, L., Domine, F., & Fily, M. (2009). Determining snow specific surface area from near-infrared reflectance measurements: Numerical study of the influence of grain shape. Cold Regions Science and Technology, 56(1), 10-17.

References by the author:
Berryman, James G. "*Relationship between specific surface area and spatial correlation functions for anisotropic porous media*." Journal of mathematical physics 28.1 (1987): 244-245.

---

## Referee Report (RR1)

**General comments**

The current version of the manuscript has been significantly reshaped since the initial submission. The authors followed most of the reviewers suggestions, which is appreciated. The overall clarity of the manuscript has been improved and critical issues are now discussed in an appropriate way.

Below I provide some complementary suggestions, mostly for the consequent parts that are entirely new at this stage. I let the authors decide whether they find those valuable or not.

Page an line numbers correspond to the changes-tracking version of the revised manuscript.

**Technical comments**

**Abstract**

The first sentence could probably be improved. I would start with something like: "At first order, specific surface area (or optical grain size) is the primary parameter used to simulate snow optical and microwave properties. However, the latter also depend on grain shape...."

I would also suggest being more explicit in the results:

1.5-8 : e.g.: "We show that the exponential correlation length, widely used for snow microwave modeling, can be expressed in terms of SSA and  $\lambda_2$ . Likewise, we show that the absorption enhancement parameter *B* and the asymmetry factor *g*, that determine snow optical properties, can be related to  $\mu_2$ ."

The last sentence is rather unclear. State that this approach allows a simultaneous understanding of snow microwave and optics. Or allows to reconciliate both fields. I would also add a more practical sentence at the end, pointing to the potential applications or suggestions for future work etc.

Generally speaking, an abstract should give as much as possible quantitative results and implications of the work. It should avoid general statements such as : "We derive relationships, we present a method, we introduce a concept...". Such statements should be placed in the introduction rather than in the abstract.

**Introduction**

Use either *correlation function* or *two-point correlation function*, but try not to alternate.

P2 l.25 : Picard et al. (2009) do not really mention *B* and *g*. They use Monte Carlo ray-tracing on different collections of geometrical shapes instead. Hence for sake of clarity, the reference should be put after "attributed to shape", rather than at the end of the sentence. It might be useful to add The Kokhanovsky and Zege (2004) reference after introducing *B* and *g*, because this is where they originate from (at least *B*).

P2 l.27 : "the question remains which..."

p5 l.17 : in terms *of the* ?"

p7 l.22 : g is the asymmetry factor. In fact I would say "(phase function, single scattering albedo, etc)" because g is just computed from the phase function.

Fig. 2 caption : parameter

p18 l.8 : are calculated ?

P20 l.18 : in the obtaining ?

P20 l.22 : I do not understand the argumentation. Why should this ratio be constant? And constant for all samples? On the contrary, I would expect  $\lambda_2$  to depend on the samples, because this is a shape parameter. To me,  $\lambda_2$  should be resolution insensitive. I would have expected you used different resolutions for the same sample and check that the retrieved  $\lambda_2$  does not change. Maybe I just did not understand well your point, but it might be useful to rephrase this part of the paragraph.

P21 l 1-2 : is there a problem with the syntax of the sentence ?

P21 l.3 : remove "is", remove "the"

p21 l.12 : a corollary is whether anisotropic media can be satisfactorily represented by "equivalent" isotropic media, for microwave and optical properties. This is probably beyond the scope of this paper, but one sentence at the end of this section 5.1.3 might be relevant if you have an opinion on this.

P25 l.2 : reference in parentheses

p25 l.18 : remove parentheses for reference

p25 l.22 : statistically

p25 l.24 : an exponential

p25 l.29 : correlation

p27 l.24 : in Libois et al. (2013)

p27 l.31 : although the range of *B* obtained experimentally is larger than that resulting from Malinka (2014), because the latter implies a shape independent *B* at weakly absorbing wavelengths, it is worth noting that the actual values are very similar, which suggests that the random two-phase medium is not inconsistent with laboratory and field measurements.

P28 l.25 : "length scales () of snow samples, which"

p30 l.11 : involves

p31 l.10 : moments

---

## Author Response (AR2)

**General comments**

The current version of the manuscript has been significantly reshaped since the initial submission. The authors followed most of the reviewers suggestions, which is appreciated. The overall clarity of the manuscript has been improved and critical issues are now discussed in an appropriate way.

Below I provide some complementary suggestions, mostly for the consequent parts that are entirely new at this stage. I let the authors decide whether they find those valuable or not.

Page an line numbers correspond to the changes-tracking version of the revised manuscript.

**Technical comments**

Abstract

The first sentence could probably be improved. I would start with something like:"At first order, specific surface area (or optical grain size) is the primary parameter used to simulate snow optical and microwave properties. However, the latter also depend on grain shape...."

I would also suggest being more explicit in the results:l.5-8 : e.g.: "We show that the exponential correlation length, widely used for snow microwave modeling, can be expressed in terms of SSA and $\lambda_2$. Likewise, we show that the absorption enhancement parameter $B$ and the asymmetry factor $g$, that determine snow optical properties, can be related to $\mu_2$."

The last sentence is rather unclear. State that this approach allows a simultaneous understanding of snow microwave and optics. Or allows to reconcile both fields. I would also add a more practical sentence at the end, pointing to the potential applications or suggestions for future work etc.

Generally speaking, an abstract should give as much as possible quantitative results and implications of the work. It should avoid general statements such as : "We derive relationships, we present a method, we introduce a concept...". Such statements should be placed in the introduction rather than in the abstract.

Reply: We agree with the second and third suggestions. The first sentence we would prefer to keep, since it emphasizes the overall topic of the paper.

The abstract is reformulated to: Grain shape is commonly perceived as a morphological characteristic of snow beyond the optical diameter (or specific surface area) which has an influence on physical properties. In this study we use tomography images to investigate two objectively defined metrics of grain shape that naturally extend the characterization of snow in terms of the optical diameter. One is the curvature-length $\lambda 2$, related to the third order term in the expansion of the two-point correlation function and the other is the second moment mu2 of the chord length distributions. We show that the exponential correlation length, widely used for microwave modeling, can be related to the optical diameter and $\lambda 2$. Likewise, we show that the absorption enhancement parameter B and the asymmetry factor g, required for optical modeling, can be related to the optical diameter and $\mu 2$. We establish various statistical relations between all size metrics obtained from the two-point correlation function and the chord length distribution. Overall our results suggest that the characterization of grain shape via $\lambda 2$ or $\mu 2$ is virtually equivalent since both capture similar aspects of size dispersity. Our results provide a common ground for the different grain metrics required for optical and microwave modeling of snow.

Introduction

Use either *correlation function* or *two-point correlation function*, but try not to alternate. Reply: We agree. We systematically use the two-point correlation function.

P2 l.25 : Picard et al. (2009) do not really mention *B* and *g*. They use Monte Carlo ray-tracing on different collections of geometrical shapes instead. Hence for sake of clarity, the reference should be put after "attributed to shape", rather than at the end of the sentence. It might be useful to add The Kokhanovsky and Zege (2004) reference after introducing *B* and *g*, because this is where they originate from (at least *B*). Reply: We agree. The suggested references are placed accordingly.

P2 l.27 : "the question remains which..." Reply: We agree. The sentence is changed accordingly.

p5 l.17 : in terms **of the** ?". Reply: We agree. The sentence is changed accordingly.

p7 l.22 : *g* is the asymmetry factor. In fact I would say "(phase function, single scattering albedo, etc)" because *g* is just computed from the phase function. Reply: We agree. The sentence is changed accordingly.

Fig. 2 caption : parameter Reply: Yes. The typo is corrected.

p18 l.8 : are calculated ? Reply: We agree, sentence is changed accordingly.

P20 l.18 : in the obtaining ? Reply: 'the' is removed to make the sentence correct.

P20 l.22 : I do not understand the argumentation. Why should this ratio be constant? And constant for all samples? On the contrary, I would expect $\lambda 2$ to depend on the samples, because this is a shape parameter. To me, $\lambda 2$ should be resolution insensitive. I would have expected you used different resolutions for the same sample and check that the retrieved $\lambda 2$ does not change. Maybe I just did not understand well your point, but it might be useful to rephrase this part of the paragraph.

Reply: The confusion might come from the fact that $\lambda_2$ is still a length scale, and not a dimensionless shape parameter. So it is naturally affected by resolution. In general the fraction of any physical length-scale and resolution should be high to sufficiently resolve that length scales by a discrete representation. By showing that the fraction of $\lambda_2$/voxelsize is approximately the same over the data set (and sufficiently large), we can exclude a resolution bias. We will explicitly stress in the discussion that our shape metrics are still length

scales in contrast to a perception of a dimensionless parameter. Sentence in 5.3.1 last paragraph is added: "Note that within this definition grain shape is not a dimensionless parameter. With this perception of shape we now…"

P21 l 1-2 : is there a problem with the syntax of the sentence ? Yes, sentence restructured to "The present dataset was previously used to study the anisotropic properties of snow ( Löwe et al. 2013). Therefore it is necessary to elaborate on the impact … "

P21 l.3 : remove "is", remove "the" We agree. Sentence is adjusted to: "Therefore it is necessary to elaborate on the impact of anisotropy in the present analysis that exclusively involves isotropic two-point correlation functions."

p21 l.12 : a corollary is whether anisotropic media can be satisfactorily represented by "equivalent" isotropic media, for microwave and optical properties. This is probably beyond the scope of this paper, but one sentence at the end of this section 5.1.3 might be relevant if you have an opinion on this.

Reply: In general it will depend on the quantity of interest if anisotropy would play a role. If the quantity of interest "performs" a similar type of volume or directional averaging the 'isotropic' approach is fine. However, commenting on these type of issues is beyond the scope of the paper and very speculative.

P25 l.2 : reference in parentheses We agree. Changed accordingly.

p25 l.18 : remove parentheses for reference We agree. Changed accordingly.

p25 l.22 : statistically We agree. Changed accordingly.

p25 l.24 : an exponential We agree. Changed accordingly.

p25 l.29 : correlation We agree. Changed accordingly.

p27 l.24 : in Libois et al. (2013) We agree. Changed accordingly.

p27 l.31 : although the range of $B$ obtained experimentally is larger than that resulting from Malinka (2014), because the latter implies a shape independent $B$ at weakly absorbing wavelengths, it is worth noting that the actual values are very similar, which suggests that the random two-phase medium is not inconsistent with laboratory and field measurements. Reply: The sentence is changed to include this consistency to "The predicted values for B (Fig. 7) are very similar to the values obtained by experiments (Libois et al., 2014) but show a smaller variation. It should be noted that, as the authors discuss, the correlation between the experimentally obtained B and shape, as defined by Fierz et al. (2009), is statistically not significant and variations might be attributed to shadowing effects relevant at higher densities."

P28 l.25 : "length scales () of snow samples, which" Reply: We agree. Changed accordingly.

p30 l.11 : involves Reply: We agree. Changed accordingly.

p31 l.10 : moments Reply: We agree. Changed accordingly.

[revised manuscript text omitted]

---

## Author Response (AR3)

Dear Guillaume Chambon,

Thank you for accepting the manuscript. Hereby we have uploaded the required correction. We have adopted the suggested sentence and we rephrased the sentence that was not clear. Attached is the track-changed file, indicating how this sentence is exactly changed.  Additionally we have changed the last sentence of the abstract to "Our results provide a common ground for the different grain metrics required for optical and microwave modeling of snow." This last change is however not highlighted in the track-changed document.

Kind regards,

Quirine Krol and Henning Löwe,
* * *

[revised manuscript text omitted]